



# Surface melting over the Greenland ice sheet from enhanced resolution passive microwave brightness temperatures (1979 – 2019)

Paolo Colosio[1], Marco Tedesco[2,3], Xavier Fettweis[4], Roberto Ranzi[1]

[1]DICATAM, Università degli Studi di Brescia, Brescia 25123, Italy
[2] Lamont-Doherty Earth Observatory, Columbia University, Palisades, NY 10964, USA
[3] NASA Goddard Institute for Space Studies, New York, NY 10025, USA
[4] Department of Geography, University of Liège, Liège 4000, Belgium

*Correspondence to*: Paolo Colosio (p.colosio002@unibs.it)

**Abstract.** Surface melting is a major component of the Greenland ice sheet (GrIS) surface mass balance, affecting sea level rise through direct runoff and the modulation on ice dynamics and hydrological processes, supraglacially, englacially and subglacially. Passive microwave (PMW) brightness temperature observations are of paramount importance in studying the spatial and temporal evolution of surface melting in view of their long temporal coverage (1979-to date) and high temporal resolution (daily). However, a major limitation of PMW datasets has been the relatively coarse spatial resolution, being
historically of the order of tens of kilometres. Here, we use a newly released passive microwave dataset (37 GHz, horizontal polarization) made available through the NASA MeASUREs program to study the spatiotemporal evolution of surface melting over the GrIS at an enhanced spatial resolution of 3.125 Km. We assess the outputs of different detection algorithms through data collected by Automatic Weather Stations (AWS) and the outputs of the MAR regional climate model. We found that surface melting is well captured using a dynamic algorithm based on the outputs of MEMLS model, capable to detect sporadic
and persistent melting. Our results indicate that, during the reference period 1979 – 2019 (1988 – 2019), surface melting over the GrIS increased in terms of both duration, up to ~4.5 (2.9) days per decade, and extension, up to 6.9% (3.6%) of the GrIS surface extent per decade, according to the MEMLS algorithm. Furthermore, the melting season has started up to ~4 (2.5) days earlier and ended ~7 (3.9) days later per decade. We also explored the information content of the enhanced resolution dataset with respect to the one at 25 km and MAR outputs through a semi-variogram approach. We found that the enhanced product
is more sensitive to local scale processes, hence confirming the potential interest of this new enhanced product for studying surface melting over Greenland at a higher spatial resolution than the historical products and monitor its impact on sea level rise. This offers the opportunity to improve our understanding of the processes driving melting, to validate modelled melt extent at high resolution and potentially to assimilate this data in climate models.



## 1 Introduction

The Greenland ice sheet (GrIS) is the largest ice mass in the Northern Hemisphere with a glaciated surface area of about 1,800,000 km$^2$, a thickness up to 3 km, and a stored water volume of about 2,900,000 m$^3$, enough to rise the mean sea level by about 7.2 m (Aschwanden et al., 2019). In this regard, estimating mass losses from Greenland is crucial for better understanding climate system variability and the contribution of Greenland to current and future sea level rise (SLR). According to data from the Gravity Recovery and Climate Experiment (GRACE) satellite mission, which records changes in

Earth's gravitational field, Greenland lost mass at an average rate of 278±11 Gt y$^{-1}$ between 2002 and 2016 (IPCC, 2019), contributing to a sea level rise of ~7.9 mm per decade. The contribution of the GrIS to sea level rise was also accelerating at a rate of 21.9±1 Gt y$^{-2}$ over the period 1992—2010 (Rignot et al., 2011) thus indicating that monitoring GrIS together with the Antarctic ice sheet is crucial to assess the impact of global warming on SLR and the global water balance (Kargel et al., 2005; 2014; Le Meur et al., 2018). Mass can be lost through surface (e.g., runoff) and dynamic (e.g., calving) processes with total

mass loss roughly split in half between the two (Flowers, 2018). Among the processes influencing the surface mass balance (SMB), i.e. difference between accumulation (Frezzotti et al., 2007) and ablation, surface melting plays a crucial role, affecting direct loss through export of surface meltwater to the surrounding oceans and though the feedbacks between supraglacial, englacial and subglacial processes and their influence on ice dynamics (e.g., Fettweis et al., 2005, 2011, 2017; van den Broeke et al., 2016; Alexander et al. 2016).

Passive microwave (PMW) brightness temperatures (T$_b$) are a crucial tool for studying the evolution of surface melting over the Greenland and Antarctica ice sheets (i.e., Jezek et al., 1993; Steffen et al., 1993; Abdalati et al., 1995; Tedesco et al., 2009; Tedesco, 2009; Fettweis et al., 2011). The PMW-based algorithms are based on the fact that the emission of a layer of dry snow in the microwave region is dominated by volume scattering (e.g., Macelloni et al., 2001); as snow melts, the presence of liquid water within the snowpack increases the imaginary part of the electromagnetic permittivity by several orders

of magnitude with respect to dry snow conditions, with the ultimate effect of considerably increasing T$_b$ (Ulaby et al., 1986; Hallikainen et al., 1987) as shown by in situ measurement campaigns (see for instance Cagnati et al., 2004). Because of the large difference between dry and wet snow emissivity, even relatively small amounts of liquid water have a dramatic effect on the T$_b$ values (e.g., Tedesco, 2009), making PMW data extremely suitable for mapping the extent and duration of melting at large spatial scales and high temporal resolution (in view of their insensitivity to atmospheric conditions at the low frequencies

of the microwave spectrum). Consequently, PMW data have been widely adopted in melt detection studies and different remote sensing techniques have been proposed in the literature (e.g., Steffen et al., 1993; Abdalati and Steffen, 1995; Joshi et al., 2001; Liu et al., 2005; Aschraft and Long, 2006; Macelloni et al., 2007; Tedesco et al., 2007; Kouki et al., 2019; Tedesco and Fettweis, 2020).

    The capability of PMW sensors to collect useful data during both day and night and in all-weather conditions allows

surface melt mapping at a high temporal resolution (at least twice a day over most of the Earth). PMW T$_b$ records are also among the longest available remote sensing continuous timeseries and an irreplaceable tool in climatological and hydrological





studies, complementing in-situ long-term observations where they are absent or too coarse. The trade-off associated with the high temporal resolution of PMW data is the relative coarse spatial resolution (historically on the order of tens of km). This can represent a limiting factor when studying surface melting as a substantial portion of meltwater production and runoff

occurs along the margins of the ice sheet, with some of these areas being relatively narrow (of the order of a few tens of kilometers or smaller, depending on the geographic position and time of the year). The use of a product with a finer spatial resolution would allow a more effective mapping of surface melting and would also allow a better comparison between in-situ measured quantities and satellite-derived estimates, reducing uncertainties in the satellite products and allowing for potential improvements to retrieval algorithms. Lastly, finer spatial resolution tools could be helpful, should they be proven effective,

in improving mapping of meltwater over ice shelves in Antarctica and improve our understanding of the processes leading to ice shelf collapse or disintegration (e.g. van den Broeke, 2005; Tedesco, 2009).

In this paper, we report our results of a study in which surface melting over Greenland is estimated making use of a recently released product developed within the framework of a NASA MeASUREs project (https://nsidc.org/data/nsidc-0630). The product contains daily maps of PMW $T_b$s generated at an enhanced spatial resolution of a few kilometers (depending on

frequency, as explained below) between 1979 and 2019. The historical gridding techniques for PMW sensors (Armstrong et al., 1994, updated yearly; Knowles et al., 2000; Knowles et al., 2006) were based on a "drop in the bucket" approach, in which the gridded value was obtained by averaging the $T_b$ data falling within the area defined by a specific pixel. In the case of the enhanced spatial resolution product, the reconstruction algorithm adopted to build the $T_b$ maps makes use of the so-called effective measurement response function (MRF; Long et al., 1998), determined by the antenna gain pattern, which is unique

for each sensor and sensor channel. This pattern is used in conjunction with the scan geometry and the integration period, allowing for "weighting" of measurements within a certain area. The approach used to generate the enhanced resolution product (called rSIR algorithm) also addresses another issue in the historical PMW dataset, which is the need for meeting the requirements of modern Earth system Data Records or Climate Data Records (CDRs), most notably in the areas of inter-sensor calibration and consistent processing methods. More details are reported in the following sections.

We divide the results of our study into two main parts: in the first part, we report the results of the cross-calibration of different passive microwave sensors over the GrIS to assure a consistent and calibrated $T_b$ time series. Specifically, we use the newly developed spatially enhanced PMW product at Ka band (37 GHz), horizontal polarization in view of its sensitivity to the presence of liquid water within the snowpack (Ulaby et al., 1986; Macelloni et al., 2005). We prefer this frequency to the ~ 19 GHz, generally used in the literature as it is less sensitive to liquid water clouds (Fettweis et al., 2011), because the

$T_b$ at Ka band are distributed at the highest spatial resolution of 3.125 km (Brodzik et al., 2018). We, then, focus on assessing whether the noise introduced by the rSIR algorithm might limit the application of the enhanced dataset to mapping surface meltwater. Then, we focus our attention on testing and assessing existing approaches to deriving melt from passive microwave data and propose an update on a recently proposed algorithm in which meltwater is detected when $T_b$ exceeds a threshold computed using the outputs of an electromagnetic model (Tedesco, 2009). We compare results from these algorithms with

estimates of surface melting obtained from data collected by automatic weather stations (AWS) and with the outputs of the



regional climate model Modèle Atmosphérique Régional (MAR; Fettweis et al., 2017). Lastly, we focus on the analysis of melting patterns and trends over study period and investigate the information content in the enhanced resolution dataset through a semi-variogram analysis.

## 2 Datasets and methods

### 2.1 Enhanced resolution passive microwave data

We use Ka band (37 GHz), horizontal polarization $T_b$ data produced within the framework of a NASA MeASUREs project and distributed at the spatial resolution of 3.125 km (Brodzik et al., 2018) over the Northern hemisphere. Specifically, we use data collected by the SMMR-Nimbus 7, the special sensor microwave/imager (SSM/I) SSM/I-F08, SSMI/-F11, SSM/I-F13 and the special sensor microwave imager/sounder SSMI/S-F17 because of its higher orbit stability

(http://www.remss.com/support/crossing-times/). Currently, the product time series begins in 1979 and ends in 2019. Data are provided twice a day, as morning (M) and evening (E) passes. Beginning and ending acquisition times for the morning and evening passes are contained within the product's metadata, together with other information. More information can be found at https://nsidc.org/data/nsidc-0630/versions/1.

Historical gridding techniques for PMW spaceborne datasets (Armstrong et al., 1994, updated yearly; Knowles et al.,

2000; Knowles et al., 2006) are relatively simplistic and were produced on grids (Brodzik and Knowles, 2002; Brodzik et al., 2012) that are not easily accommodated in modern software packages. In the reconstruction algorithm used for the enhanced $T_b$s, the so-called effective measurement response function (MRF), determined by the antenna gain pattern and being unique for each sensor and sensor channel, is used in conjunction with the scan geometry and the integration period. The gridding approach uses the Backus-Gilbert technique (Backus and Gilbert, 1967; 1968), a general method for inverting integral

equations, which has been applied for solving sampled signal reconstruction problems (Caccin et al., 1992; Stogryn, 1978; Poe, 1990) for spatially interpolating and smoothing data to match the resolution between different channels (Robinson et al., 1992), and improving the spatial resolution of surface brightness temperature fields (Farrar and Smith, 1992; Long and Daum, 1998). More information about the product can be found at https://nsidc.org/data/nsidc-0630. An example of $T_b$ maps at 37 GHz, horizontal polarization, in the case of both the coarse and enhanced resolution products over Greenland on 16 July 2001

is reported in Figure 1. The higher detail captured by the enhanced spatial resolution is clearly visible, especially along the ice sheet edges, where melting generally occurs at the beginning of the season and lasts for the remaining part of the summer. Figure 2a shows an example of time series of both coarse (blue) and enhanced (red) PMW $T_b$ (again at 37GHz, horizontal polarization) for the pixel containing the Swiss Camp station (69° 34' 06" N, 49° 18' 57" W as shown in Figure 1). From the figure we observe that the two timeseries are highly consistent with each other, with a mean difference of 0.895 K and standard

deviation of 4.89 K, indicating that the potential noise introduced by the enhancement process is not a major issue. Yet, differences do exist, as in the case of 3 April 2012 (DOY 93), when the enhanced product suggests the presence of melting while the coarse product does not (Figure 2). This is likely due to the different spatial resolution between the two products, as



we discuss in the following sections and shows the added value of using the 37 GHz frequency in detecting small scale features of the melting process.

## 2.2 Greenland air/surface temperature data

In order, to assess the results obtained from PMW data, we use in-situ data collected by AWS distributed over the Greenland Ice sheet. In the absence of direct observations of melting, we use air temperature (3 m above the surface) to extrapolate instances when liquid water is present, following the procedure adopted by Tedesco (2009) for Antarctica. Specifically, we use data recorded by stations of the Greenland Climate Network (GC-Net; Steffen et al., 1996). The AWSs provide continuous measurements of air temperature, wind speed, wind direction, humidity, pressure and other parameters. We focus on air temperature data collected every hour by 17 selected stations reported in Table 1. We considered a validation period from 2000 (when all the considered AWS were in operation) to 2016 and used daily averaged values. More information about the GC-Net dataset can be found at http://cires1.colorado.edu/steffen/gcnet/.

## 2.3 The MAR model

We assess the enhanced PMW-based surface melt maps with the outputs of the regional climate model Modèle Atmosphérique Régional (MAR, e.g., Alexander et al., 2014; Fettweis et al., 2013; Fettweis et al., 2017; Tedesco et al., 2013). MAR is a modular atmospheric model that uses the sigma-vertical coordinate to simulate airflow over complex terrain and the Soil Ice Snow Vegetation Atmosphere Transfer scheme (SISVAT) (e.g., De Ridder and Gallée, 1998) as the surface model. MAR outputs have been assessed over Greenland (e.g., Fettweis et al., 2005; Fettweis et al., 2017; Alexander et al., 2014). The snow model in MAR, which is based on the CROCUS model of Brun et al. (1992), calculates albedo for snow and ice as a function of snow grain properties, which in turn depend on energy and mass fluxes within the snowpack. Lateral and lower boundary conditions are prescribed from reanalysis datasets. Sea-surface temperature and sea-ice cover are prescribed using the same reanalysis data. The atmospheric model within MAR interacts dynamically with SISVAT.

In this study, we use the output from MAR version v3.11.2 characterized by an enhanced computational efficiency and improved snow model parameters (Fettweis et al., 2017; Delhasse et al., 2020). The model is forced at the boundaries using ERA5 reanalysis (Hersbach et al., 2020), the newest generation of global atmospheric reanalysis data that superseded ERA-Interim (Dee et al., 2011), and output is produced at a horizontal spatial resolution of 6 km. In order to compare output from MAR with estimates of meltwater extent obtained from PMW data, we average the LWC simulated by MAR along the vertical profile, following Fettweis et al. (2007).

## 2.4 Melt detection algorithms

Generally speaking, melt detection algorithms can be divided into threshold-based (T-B) and edge-detection (E-D) algorithms (e.g., Liu et al., 2005; Joshi et al., 2001; Steiner and Tedesco, 2014). Here we focus on threshold-based algorithms, detecting melting when $T_b$ values (or their combination) exceed a defined threshold, computed in different ways depending on



the algorithm. For example, Steffen et al. (1993) used the normalized gradient ratio $GR=(T_{b19H}-T_{b37H})/(T_{b37H}+T_{b19H})$ to detect
wet pixels with a threshold value computed based on in-situ measurements. This method was later updated by Abdalati and
Steffen (1995) who introduced the cross-polarized gradient ratio $XPGR=(T_{b19H}-T_{b37V})/(T_{b37V}+T_{b19V})$, where the Ka-band
component of the algorithm was switched from horizontally to vertically polarized.

Aschraft and Long (2006) proposed a threshold based on dry (winter) and wet snow $T_b$ as $T_c=\alpha T_{winter}+(1-\alpha) T_{wet}$
where $T_{winter}$ is the average of winter $T_b$ and $T_{wet}$ fixed as 273 K. The mixing coefficient $\alpha=0.47$ was derived considering
LWC=1% in the first 4.7 cm of snowpack. Similarly, a method based on a fixed threshold (set to 245 K and derived from the
outputs of electromagnetic model) above which melting is assumed to be occurring was proposed in (Tedesco et al., 2007).

Several other studies have been detecting melting when $T_b$ values exceed the mean winter value $T_{winter}$ plus an
additional value $\Delta T$ (M+$\Delta T$ approaches, where M represents the January-February mean brightness temperature) associated
with the insurgence of liquid water within the snowpack:

$$T_c = T_{winter} + \Delta T, \tag{1}$$

Torinesi et al. (2003) proposed a value of $\Delta T=N\sigma$ with $T_{winter}$ and $\sigma$ varying in space and time but fixed N=3 from the analysis
of weather station temperature data. Zwally and Fiegles (1994) used a fixed value of $\Delta T=30$ K. Tedesco (2009) proposed an
alternative approach based on the outputs of the Microwave Emission Model of Layered Snowpack (MEMLS) electromagnetic
model (Weisman and Matzler, 1999). In this case, an ensemble of outputs is generated by MEMLS by varying the inputs (e.g.,
correlation length, LWC, density, etc.). These outputs are, then, used to build a linear regression model for the $\Delta T$ that is a
function of the winter $T_b$ value as follows:

$$\Delta T = \varphi T_{winter} + \omega, \tag{2}$$

with the values of the coefficients obtained from the linear regression. This is done to account for the increment related to the
presence of LWC within the snowpack as a function of the snow properties: a fixed increase would correspond to different
values of LWC, potentially making the mapping of the wet snow areas inconsistent in terms of LWC values. For example: a
snowpack with small grain size will require a relatively larger amount of LWC with respect to a snowpack with larger grain
size for the $T_b$ values to increase by 30 K. Or, from a complementary point of view, an increase of 30 K due to presence of
liquid water in the case of a snowpack with relatively coarse grains will correspond to a lower value of LWC than an increase
occurring in a snowpack with smaller grain size. In summary, the adoption of this approach provides consistency in terms of
the minimum LWC that is detected by the algorithm. Building on Tedesco (2009), we considered the two LWC values of 0.1
% and 0.2 %. The coefficients are $\varphi = -0.2$ and $\omega = 58$ K ($R^2=0.91$) in the case of LWC=0.1% and $\varphi = -0.52$ and $\omega = 128$
K ($R^2=0.92$) in case of LWC=0.2%. The $T_b$ threshold value computed in this case can, therefore, be written as follows:

$$T_c = T_{winter} + \varphi T_{winter} + \omega = (1 + \varphi)T_{winter} + \omega = \gamma T_{winter} + \omega, \tag{3}$$

where ($\gamma$, $\omega$) assume the values of 0.80 and 58 K in the case of LWC=0.1% and 0.48 and 128 K in the case of LWC=0.2%.





Here, we focus five approaches: the M+ΔT approach choosing ΔT equal to 30K, 35K and 40K as sensitivity to Zwally and
       Fiegles (1994) (M+30, M+35 and M+40 from here on), the algorithm based on MEMLS (MEMLS from here on) and the 245
       K fixed threshold (245K from here on). In the following sections, we report the results of two algorithms, namely the one using
       a fixed threshold of 245 K and the one based on MEMLS (0.2 % LWC). As we explain below, this choice was driven by the
       performance of the different considered algorithms. Moreover, we found that the fixed-threshold algorithm is more sensitive

to persistent melting where the MEMLS-based one can detect sporadic melting. This allows us to analyze both melting
       conditions (sporadic vs. persistent) and analyze them within the long-term, large spatial scales that the PMW data can provide.

**3 Inter-sensor calibration of enhanced resolution passive microwave data**

       In view of the novel nature of the PMW dataset, we first focus on the cross-calibration of the data acquired by the
       different sensors. This initial processing step aims to account for biases and differences associated with swath width, view

angle, altitude and Local-Time-Of-Day (LTOD) as well as the specific intrinsic differences associated with each sensor on the
       different platforms (Table 2). Several approaches have been proposed in the literature to address this issue for the historical,
       coarser spatial resolution gridded datasets. For example, Jezek et al. (1993) compared SMMR and SSM/I over the Antarctic
       ice sheet for K and Ka bands (~ 19 GHz and ~ 37GHz) for both horizontal and vertical polarizations. Steffen et al. (1993)
       proposed an approach focusing over Greenland for the K-band; Abdalati et al. (1995) derived relations between SSM/I

observations for the F08 and F11 platforms over Antarctica and Greenland for 19.35GHz, 22.2GHz and 37 GHz. Dai et al.
       (2015) intercalibrated SMMR, SSM/I (F08 and F13) and SSMI/S (F17) over snow covered pixels in China and SMMR, SSM/I
       and AMSR-E over the whole Earth surface sampling hot and cold pixels.

       Given the novelty of the $T_b$ products used here and the absence of specific intercalibration of data collected from
       different platforms for this product, we developed an ad-hoc intercalibration for the enhanced PMW dataset. Following Stroeve

et al. (1998), we perform the intercalibration using only data collected over the Greenland ice sheet. The overlapping periods
       for the different sensors are the following: SMMR and SSM/I-F08 overlap between 07/09/1987 and 08/20/1987 for a total of
       22 days (one every two days as sensed by SMMR sensor); F08 and F11 overlap between 12/03/1991 and 12/18/1991 for a
       total of 16 days, F11 and F13 overlap between 05/03/1995 and 09/30/1995 for a total of 76 days; and F13 and F17 overlap for
       the period 03/01/2008-12/10/2008 for a total of 71 days. We perform a linear regression between the data acquired by two

sensors over the Greenland ice sheet and calculate the slope (m) and intercept (q) of the linear regression

$$y = mx + q, \qquad\qquad\qquad (4)$$

In Eq. (4) x and y represent the $T_b$ values from coincident data from the two overlapping satellite products. We consider two
approaches to compute the m and q values in Eq. (4). In the first method we compute the weighted average of the daily slope
and intercept values from the regression of daily data. Considering n days, for every i-th day we first compute $m_i$, $q_i$ and the





coefficient of determination for the linear regression of Eq. (4) ($R^2_i$), and then we average them according to Eq. (5) and Eq. (6).

$$m = \frac{\sum_{i=1}^{n} m_i R_i^2}{\sum_{i=1}^{n} R_i^2},$$ (5)

$$q = \frac{\sum_{i=1}^{n} q_i R_i^2}{\sum_{i=1}^{n} R_i^2},$$ (6)

This choice assigns higher values to the weights obtained from pairs of data with higher correlation. In the second
method, we consider all values for all days when data from both platforms are available and then evaluate m and q through a linear regression fitting procedure based on least-square fitting. As an example, in Figure 3 we show the scatter plots of the data used for the linear regression for Greenland for both evening and morning passes for the SMMR and SSM/I-F08 sensors, reporting values of m, q and $R^2$. Using the estimated values of m and q, we then correct the values for one of the sensors by applying Eq. (4) to the $T_b$ values of one sensor (x, e.g. SMMR) to obtain new corrected $T_b$ values (y).

We perform an additional comparison using the average difference between the brightness temperature values and evaluating the matching between histograms of the overlapping data (Dai et al., 2015) by means of the Nash-Sutcliffe Efficiency (NSE) coefficient (Nash and Sutcliffe, 1970), defined as:

$$NSE = 1 - \frac{\sum \left( h_i(T_b^A) - h_i(T_b^B) \right)^2}{\sum \left( h_i(T_b^B) - \bar{h}(T_b^B) \right)^2},$$ (7)

where $h_i$ is the absolute frequency of the i-th value of brightness temperature of the two sensors (A and B) considered. The
NSE is usually applied in calibration/validation procedures to assess the matching between measured and modelled quantities, as in Subsection 4.2. After the application of linear relations found using Eq. (4) through (6), in order to quantitatively assess the impacts of the intercalibration on $T_b$ values, we computed the absolute difference between the values of the histograms of the $T_b$s obtained as:

$$D_i = \left| h_i(T_b^A) - h_i(T_b^B) \right|,$$ (8)

where $D_i$ is the absolute difference between the two histograms A and B for the i-th value of brightness temperature. Then, we sum the differences over the total number of pixels and compute the relative variation as follows:

$$d = \frac{D_{original} - D_{corrected}}{D_{original}},$$ (9)

where $D_{original}$ and $D_{corrected}$ are, respectively, the summations of $D_i$ before and after the calibration. The relative variation d can range from $-\infty$, indicating worsening in matching of the histograms, to 1, indicating a perfect matching of the histograms after
the intercalibration.

We point out that the overlap between SMMR and SSM/I-F08 data occurs in the months of July and August. During these months, the differences between acquisition times might lead to biases and errors associated with snow conditions (e.g.,





wet vs. dry). We report in Table 3 average values of the difference between pairs of $T_b$ data and values of the NSE coefficient

for the histograms of the same pairs. In Table 4 we report the values for slope and intercept obtained from the linear regression

analysis of enhanced PMW $T_b$s (37 GHz, H-pol.) over Greenland for SMMR vs. SSM/I-F08, F08 vs. F11, F11 vs. F13 and

F13 vs. F17, together with the $R^2$ values and values of d computed according to Eq. (6). In the case of SSM/I and SSMI/S, R2

values are higher, mostly around 0.98. In Figure 4 we also show examples of histograms in the case of the SMMR and SSM/I

F08 sensors. Large differences are obtained in the case of the SMMR and SSM/I-F08, for both evening and morning passes,

likely because of the difference in overpass time and the presence/absence of melting in some of the scenes observed by one

sensor but not present in the other. On the other hand, in the case of the SSM/I and SSMI/S sensors, the average difference is

close to 0 K (with the exception of the F-08 and F-11 satellites showing an average difference slightly larger than 1 K,

consistent with previous results obtained by Abdalati et al. (1995) in the case of the 25 km resolution data) together with NSE

values extremely close to 1 (Table 3). Still in the case of SMMR and SSM/I-F08, the higher average difference (ranging

between -3.4 K to -4.3 K) and the relatively lower NSE values (ranging between 0.89 and 0.96) show that these sensors show

the largest bias. Lastly, we only applied the correction to SMMR and we did not apply the linear regression to the SSM/I F08

– SSM/I F11, SSM/I F11 – SSM/I F13, SSM/I F13 – SSMI/S F17 datasets as, in this case, the linear correction worsened the

agreement between the two sets of measurements.

## 4 Results and discussion

### 4.1 Assessment of melt detection algorithms

265        In order to assess the capability of the selected algorithms, we compare the outputs obtained by PMW data with in

situ air/surface temperature as an index of surface melting (Braithwaite and Oelsen, 1989) daily averaged from AWS and with

the liquid water content simulated by the regional climate model MAR. We first evaluate performances at local scale (at the

specific locations of the selected AWS), comparing the number and the concomitance of melting days according to PMW and

the ground truth reference. Then, according to the results obtained, we focus on MEMLS and 245K algorithms to evaluate at

ice sheet scale the capability of the two approaches in describing the surface melt extent.

### 4.1.1 Assessment with AWS data

        Historically, the presence of liquid water within the snowpack using data from AWS has been estimated when

recorded air temperature exceeds a certain threshold.  Because melting can also occur because of radiative forcing (i.e., solar

radiation) and the air temperature does not necessarily represent the snow surface temperature, we tested three threshold values

for air temperature of  0ºC, -1°C and -2°C, as in Tedesco (2009). We assessed the performance of the PMW-based algorithm

by defining commission and omission errors. Commission error occurs when melting is detected by PMW data but not by

AWS data and omission error occurs when melting is detected by AWS data but not by PMW. The results of the error analysis

are summarized in Table 5 in the case of the different algorithms and for the different threshold values on the AWS air



temperature values. In the table, values of commission and omission errors are reported as an average over all stations. Specific

results for each AWS location are reported in the supplementary material. Our results indicate that the 245K algorithm shows the lowest commission error (between 0.31% and 0.63%) and the highest omission error (5.38%-919%). This is consistent with this algorithm being the most conservative among those considered (i.e., the algorithm is not sensitive to sporadic melting). In contrast, a higher commission error is achieved in the case of the M+30, M+35 and M+40 thresholds, particularly for the Humboldt and GITS stations (North-West Greenland), where the commission error is up to one order of magnitude

larger than in the case of MEMLS and 245K algorithms for every ground-truth reference (e.g. from 0.70% for MEMLS and 0.09% for 245K to 5.63% for M+30, in case of $T_{air}=0°$). Moreover, we note in the case of the MEMLS algorithm the lowest omission error in Swiss Camp, JAR-1 and JAR-2 sites (6.9% for MEMLS and 8.4% for M+30, 10.0% for M+35, 12.4% for M+40 and 17.4% for 245K). The sensitivity to the air/surface temperature threshold is low, with commission and omission error, respectively decreasing by about 1% and increasing by 3% when considering threshold values from 0°C to -2°C.

290          In order to better understand the sources of the relatively high values of the commission errors at some locations, we show in Figure 5 the timeseries of surface air temperature and $T_b$ at 37 GHz H-pol. at three selected stations: a) Summit, b) Humboldt and c) Swiss Camp for the year 2005. The threshold values obtained with the different detection algorithms are also plotted as horizontal lines (black) as well as the 0°C air/surface temperature threshold (magenta). We selected these three locations as they are representative of three different environmental and melting conditions. The timeseries recorded at Summit

station (Figure 5a) shows the sensitivity of brightness temperature to physical temperature and its seasonal variations. In this case, the surface/air temperature remains below 0°C throughout year and the $T_b$ signal does not exceed any threshold value (horizontal lines). This timeseries is typical of a location where melting is generally absent. The $T_b$ timeseries collected in correspondence of Humboldt location (Figure 5b) shows a strong and sudden peak starting on 20 July, when the air/surface temperature average is about -0.5 °C (detected by -1°C and -2°C air temperature thresholds). This event is detected by all

algorithms. Nevertheless, the M+30 (and similar algorithms) indicate the potential presence of melting also for the period preceding the July melting (between 17 June and 17 July). This melting is not confirmed by other algorithms or by the AWS analysis, suggesting that the threshold value used for these algorithms might be too low. Lastly, melting clearly occurs in the case of Swiss Camp (Figure 5c), characterized by the sharp and substantial increment of $T_b$ beginning around mid-May. For this case, all algorithms detect melting, with the MEMLS being the most sensitive and the 245K fixed threshold being the most

conservative. The computed rough estimation of the average emissivity for the period 17 June – 17 July (as Tb divided by the recorded surface temperature) also suggests that melting is not occurring in the considered period in Humboldt case, presenting an average emissivity even lower than in Swiss Camp case. Figure 6 shows maps of surface melt extent obtained using the different approaches for July 13th, 2008. Consistent with the results discussed above, the M+30K and M+35K algorithms suggest melting up to high elevations, within the dry snow zone, where it likely did not occur. The M+40K and MEMLS

algorithms show similar results, while the 245K fixed-threshold approach shows, as expected, the most conservative estimates. As mentioned, the threshold algorithms for $\Delta T_b$ (M+30, etc.) rely on a fixed $\Delta T_b$ value, which could produce errors if there is a large seasonal range in brightness temperatures due to temperature variability. In contrast, the MEMLS algorithm is based





on the linear regression of the $\Delta T_b$ as function of different combinations of dry snow conditions (LWC=0, i.e. different winter brightness temperature means). This provides an appropriate threshold value that takes into account the snow conditions before

melting and, at the same time, follows a more consistent approach with respect to the amount of LWC detected in the snowpack.

### 4.1.2 Assessment with MAR outputs

For the comparison between PMW-based and MAR outputs, we averaged the vertical profiles of LWC computed by MAR to the top 5 cm ($MAR_{5cm}$) and the top 1m ($MAR_{1m}$) of snowpack following Fettweis et al. (2007). In order to be consistent with the minimum LWC to which the MEMLS algorithm is sensitive, we set the threshold on the LWC values to which we

assume melting is occurring to 0.2% for both depths. We selected two different depths for our analysis so we could study two types of melting events: (1) sporadic surface melting, affecting the first few centimeters of the snowpack, and (2) persistent subsurface melting, affecting the snowpack from the surface up to around the first meter. For consistency with the AWS analysis, we report the results averaged over those MAR pixels containing the AWS stations discussed in the section above in Table 5. The comparison between the results obtained from the PMW and modelled LWC indicates that the more conservative

approaches (i.e., 245K) perform better when considering the LWC averaged on the first 1m of snowpack. In fact, the 245K threshold shows the lowest overall error for this case (C+O = 6.06%). In the case of the top 5cm, all the algorithms present similar performances on average, with the best performance obtained again in the case of the 245K (C+O=5.53%). However, all the $M+\Delta T_b$ algorithms present the same issue of larger commission error than MEMLS and 245K (e.g., from 0.99% for MEMLS and 0.26% for 245K to 4.62% for M+30) in North-East Greenland (e.g., Humboldt and GITS stations). This confirms

the results we obtained from the comparison with AWS data, pointing out the overestimation of melting in some dry areas by $M+\Delta T_b$. For both the $MAR_{1m}$ and $MAR_{5cm}$ cases, for all the considered algorithms, we find a high commission error in the cases of the JAR-1, JAR-2 and Swiss Camp sites (between 10% and 22%).

In order to better understand the origins of these errors, we show in Figure 7 further insights into the differences between the PMW brightness temperature and MAR outputs. Figure 7a and b show, respectively, the timeseries of LWC

averaged over the first 5cm (LWC5cm) and 1m (LWC1m) obtained from MAR at the Swiss Camp site. In Figure 7c we report the $T_b$ timeseries and the daily average surface temperature (threshold values reported as horizontal lines). We first note an early melt event (labeled LWC=0.046% in Figure 7b for the 108th day of the year) detected by PMW MEMLS algorithm and at the AWS station but apparently undetected in $MAR_{1m}$. A closer look to the time series shows that in fact $MAR_{1m}$ does estimate a LWC of 0.046% on this day while $MAR_{5cm}$ a LWC of 0.6%. This suggests that in some cases (before the main melt

season) the MEMLS algorithm is actually sensitive to the LWC in the first 5 cm of snowpack, as a consequence of the approximation of the electromagnetic outputs imposed by the linear fitting. We also note a melt event (labeled with LWC=0.5% in Figure 7b) at the end of the melting season detected by both AWS data and MAR (both in the first 5cm and 1m of snow) but not by any PMW algorithm. The $T_b$ timeseries reveals a small peak, but the signal is not strong enough to exceed any threshold. This correspond to a rainfall event (simulated by MAR and accompanied by values of measured relative humidity

reaching saturation during the day) suggesting that the sensitivity to liquid clouds of the 37 GHz channel could mask some



melt events (Fettweis et al., 2011). Moreover, at the end of the melting season the brightness temperature appears to be slightly lower than January/February average, possibly because of an increment in grain size after refreezing, leading to a lower emissivity.

The results discussed above (together with results from the comparison with AWS data) suggest that 245K is the most conservative among the approaches we tested, providing the lowest (highest) commission (omission) error but being unable to detect sporadic melt events. On the contrary, the MEMLS and M+$\Delta T_b$ algorithms can detect sporadic melt events and present lower omission error compared to 245K. However, M+$\Delta T_b$ algorithm overestimate melting in some dry areas (North West of the GrIS), suggesting melting when it is not actually occurring. Contrarily, MEMLS algorithm is not affected by the large commission error in dry areas, presents the lowest omission error in Swiss Camp area (together with M+30) and is still sensitive

to low levels of LWC. Considering the overall error (C+O Mean in Table 5), the MEMLS algorithm shows the best performance (6.66%). In view of the presented analysis and the different sensitivity to surface and subsurface melting, in the following we focus on the 245K and MEMLS algorithms to study the extent of persistent and sporadic surface melting, respectively.

As a further analysis, we compared the PMW-retrieved melt extent (ME) with that estimated from MAR outputs. In

Figure 8 we show the timeseries of melt extent integrated over the whole ice sheet for two selected years ( (a) 1984 and (b) 2006, selected randomly to present an example of SMMR and SSM/I cases) estimated from the values of MAR LWC averaged along the first 5 cm and 1m of snowpack together with the timeseries of the melt extent from the PMW data. For each year, we compute the daily melt extent for the period 1 May to 15 September and use the Nash-Sutcliffe Efficiency (NSE) coefficient (Nash and Sutcliffe, 1970), described in Section 3, for a quantitative analysis. We remind here that NSE can assume values in

the interval (-∞,1]. A perfect match between the two timeseries is achieved when the NSE value is 1. Values of NSE in the [0,1] interval indicate that the modelled variable is a better predictor of the measurements than the mean. If NSE is a negative number, the mean of the measured data describes the timeseries better than the modelled predictor. Here, we chose NSE=0.4 as efficiency threshold, considering that we compute melt extent at daily timescale and from datasets at two different resolutions (i.e., resulting in an intrinsic bias related to the different pixel size). We compared the timeseries of melt extent

(ME) obtained using the 245K algorithm with $MAR_{1m}$ (245K vs. $MAR_{1m}$) and the MEMLS melt extent with $MAR_{5cm}$ (MEMLS vs. $MAR_{5cm}$). We report NSE coefficients computed for the 41-year period in Table 6. At first, we notice that for the 1979-1992 period the comparison between 245K and $MAR_{1m}$ produces large negative NSE values, indicating an unsatisfactory match between PMW and MAR derived melt extents. The comparison between MEMLS and $MAR_{5cm}$ presents negative values of NSE as well (unsatisfactory results). Between 1987 and 1992, we found larger but still negative NSE values presenting

smaller absolute values. Between 1993 and 2019, we found for every year negative values of NSE for 245K and positive values for MEMLS, indicating satisfactory results only for the latter algorithm. The timeseries in Figure 8a reveals a strong underestimation of 245K-derived melt extent relative to $MAR_{1m}$ (the cause of low NSE= -151.596) and shows the slightly better matching in case of MEMLS (NSE=-0.540). This result suggests that in case of SMMR data brightness temperature values cannot always reach the 245 K threshold, even if the snowpack is saturated with liquid water and surface melting is





developed. As a consequence, the 245K threshold might be too high in the first part of the dataset, resulting in an underestimation of the melt extent. On the contrary, MEMLS threshold, generally lower, can better capture the spatiotemporal evolution of surface melting, even if the melt extent is still underestimated. In Figure 8b, the timeseries' obtained with 245K appears to better follow the temporal variability of melt extent from MAR during the melting season but still presenting a strong underestimation (NSE= - 5.250). On the other hand, MEMLS-derived timeseries better matches the MAR-derived one,

showing a largely satisfactory Nash-Sutcliffe Efficiency coefficient (0.782).

In summary, 245K threshold, even if presenting acceptable results in terms of commission and omission error considering both AWS and MAR comparison, is too high to fully capture the melt extent everywhere over the ice sheet. Contrarily, we found that MEMLS algorithm is suitable in capturing the evolution of melting over the Greenland ice sheet.

## 4.2 Surface melting trends

Here, we report results concerning trends of melt duration, length of the melting season and melt extent. We define melt duration (MD) as the total number of days when melting is detected. We compute trends of MD over the whole ice sheet (mean melt duration, MMD, averaged over the GrIS area) and at a pixel by pixel scale. We also study the maximum melting surface (MMS, maximum extent of melting area, i.e. the sum of the pixel areas in which melting has been detected at least once over a period, expressed as a fraction of the total GrIS area) and the cumulative melting surface or melting index (MI,

the sum of the melting pixel days multiplied by the area subjected to melting, i.e. the integral of the MD timeseries; Tedesco et. al, 2007). Lastly, we define melt onset date (MOD) and melt end date (MED) as the first day when melting occurs for two days in a row (MOD) and when melting does not occur for at least 2 days in a row. We report in Figure 9 the timeseries of annual values of MMD, MMS and MI for the 1979-2019 and 1988-2019 reference periods. We decided to look at two different reference periods in view of the fact that SMMR data is collected every other day and the SMMR and SSM/I sensors and that

the SMMR and SSM/I sensors are fundamentally different from each other (where this is not true in the case of the remaining SSM/I sensors). We show the results obtained applying the MEMLS algorithm (the one that presented the highest performances in all the considered cases) and the 245K threshold (because it presents good performances in the omission/commission errors analysis, even if it shows the limit of strong underestimation of melt extent from the comparison with MAR outputs). In the case of MMD (Figure 9a), we obtain a positive statistically significant (p-value<0.05) trend from both the 245K and MEMLS

algorithms (except for the 245K$_{1988-2019}$), being 0.249 d y$^{-1}$ (0.108 d y$^{-1}$) in the case of the 245 K algorithm for the period 1980 – 2016 (1988 – 2016) and 0.451 d y$^{-1}$ (0.291 d y$^{-1}$) in the case of the MEMLS algorithm. On average, these trends obtained for the MMD considering the high resolution dataset are 5.8% larger than in case of coarse resolution. Also for the maximum melting surface (MMS, Figure 9b), both the 245K and MEMLS algorithm indicate statistically significant positive trends (p-value<0.05 for every case, p-value<0.1 for MEMLS$_{1988-2019}$ ). The computed trends suggest that the MMS has been increasing

by 0.69% y$^{-1}$ in the case of MEMLS and 0.94% y$^{-1}$ in the case of the 245K algorithm for the period 1979-2019 (percentage with respect to the whole GrIS surface area). For the 1988 – 2019 period, we also found that the trends are statistically significant but smaller in value (0.36% y$^{-1}$ for MEMLS and 0.47% y$^{-1}$ for 245K). The obtained trends are on average 3.4%





lower than the ones computed using the 25 km dataset. In the case of MI (Figure 9c), we also found positive statistically significant trends of $9.166 \times 10^5$ km$^2$ d y$^{-1}$ (MEMLS) and $5.862 \times 10^5$ km$^2$ d y$^{-1}$ (245K) for the complete timeseries. When considering the reduced reference period, we found a 95% statistically significant trend of $5.726 \times 10^5$ km$^2$ d y$^{-1}$ only in case of MEMLS. The increase of the trends passing from the 25 km to the 3.125 km resolution dataset is on average equal to 10.8%. Lastly, we report in Figure 9d the melt onset date and melt end date averaged spatially over the pixels with 95% significant trends in Figure 11. We found that average MOD (crosses in Figure11d) presents similar trends for 245K and MEMLS considering both the entire and shortened time series equal to -0.546 d y$^{-1}$ and -0.273 d y$^{-1}$, respectively, in case of 245K (44% larger than in case of 25 km data) and -0.404 d y$^{-1}$ and -0.254 d y$^{-1}$ in case of MEMLS (17% larger than in case of 25 km data). On the contrary, in case of average MED (in red) we found larger differences when considering the reduced timeseries with results equal to 0.687 d y$^{-1}$ for $245K_{1979-2019}$, 0.708 d y$^{-1}$ for $MEMLS_{1979-2019}$ and 0.396 d y$^{-1}$ for $MEMLS_{1988-2019}$ (14%, 7.5% and 29% larger than in case of 25 km data). The $245K_{1988-2019}$ case does not present a statistically significant trend. This difference suggests that 245K algorithm may have stronger limitations in capturing the last portion of the melting season in case of SMMR data thus confirming the problems observed for the melt detection with this source of data vs. $MAR_{1m}$ simulations.

In Figure 10 we show the trends of melt duration (MD), melt onset date (MOD) and melt end date (MED) on a pixel-by-pixel basis for the complete time series (1979-2019). We found that the trend in melt duration exhibits the highest statistical significance (in terms of number of statistically significant pixels), being the most stable and reliable trend among the pixel-by-pixel parameters analyzed. We found mostly positive trends in melt duration in all pixels (Figure 10a and b), with higher values moving towards the coastline, maxima in the ablation zone of the Jakobshavn Isbrae (2.40 d y$^{-1}$) for 245K and 2.66 d y$^{-1}$ for MEMLS) and minima in high altitude areas. We averaged the statistically significant trends, finding an average of 0.468 d y$^{-1}$ for the 245K algorithm, and of 0.697 d y$^{-1}$ in the case of the MEMLS. In case of MOD and MED, we found a lower number of statistically significant pixels. The statistically significant pixels exhibit a negative trend for MOD (Figure 10c and d) and a positive trend for MED (Figure 10e and f), with the melting season starting on average 0.694 d y$^{-1}$ earlier and ending 0.680 d y$^{-1}$ later according to 245 K algorithm (0.360 d y$^{-1}$ earlier and 0.909 d y$^{-1}$ later for MEMLS). We point out that the average of the statistically significant trends is generally higher than the trends computed at ice sheet scale since we computed the average over the statistically significant pixels only.

## 4.3 Spatial information content

In order to investigate the spatial information content of the enhanced resolution PMW data with respect to the coarser one, we also performed a variogram-based analysis of melt duration estimated from the two products when using either the 245K or the MEMLS algorithms. Variogram analysis is generally adopted in geostatistical analyses to evaluate autocorrelation of spatial data (Delhomme, 1978; Edward et al., 1989) with variograms being characterized by three parameters: the sill, the range and the nugget effect. The sill is the variance value at which the variogram becomes flat. The range is the distance at which the variogram reaches the sill. Beyond this value, the data are no longer autocorrelated. The nugget effect is the variance



value at null distance, theoretically zero and resulting from measurement errors or highly localized variability. We point out that knowledge of scales is imperative for improving our understanding of the observed changes because processes and related relationships change with scale. Moreover, quantifying the variability of processes across scales is a critical step, ultimately leading to proper observation and modeling scale resolution. In this regard, the relationship between processes, observation

and modeling scales controls the ability of a tool to detect and describe the constituent processes. Here, we show our preliminary results of a variogram-based analysis applied to melt duration estimated from the MEMLS and 245K algorithms for the months of May through August of 2012 when using either the enhanced or the coarse resolution products. We plan to extend this analysis to other variables and periods as a part of our future plans. We also performed the same analysis applied to melt duration estimated using LWC modelled with MAR, according to the same rationale described in the previous sections.

The results of our analysis are summarized in Figure 12, where we show the empirical (blue crosses) and modelled (red line) semi-variograms for Greenland melt duration computed applying the MEMLS and 245K algorithms to both 25 km and 3.125 km resolution data for the months of May through August of 2012, and in Table 7, where we report the parameters of the spherical fitting of the empirical semi-variogram in case of melt duration obtained according to $LWC_{1m}$ and $LWC_{5cm}$ approaches. At first, we note that $R^2$ values of the fitting for the modeled variograms are consistently higher in the case of

enhanced resolution data, suggesting that enhanced resolution data might be more suitable for a variogram-based analysis. For the coarse resolution data, we found $R^2$ values of comparable magnitude with the enhanced resolution case only in May. When computing the spherical fitting of the empirical variograms of melt duration from MAR, we found for each case considered similar $R^2$ values (between 0.118 and 0.484). The values for the range in the case of the 3.125 and 25 km products are similar in May for the 245K algorithm (on the order of ~ 200 km) but they appear different in the case of the MEMLS algorithm, when

the enhanced product shows a lower value of ~ 170 km against ~ 270 km in the case of the coarse product. This could be due to the fact that the MEMLS algorithm is more sensitive to sporadic melting and when applied to the enhanced $T_b$ dataset it allows to detect melting driven by processes whose scale cannot be captured by the coarser nature of the historical dataset. In case of MAR, the value of the range results lower in case of $LWC_{5cm}$ (187.70 km) than in case of $LWC_{1m}$ (199.17 km), suggesting again the affinity of MEMLS algorithm with melting strictly confined in the very first layer of snowpack. As the

melting season progresses, the variograms of the coarse dataset shows either similar values for the range or a poor fit of the experimental variogram. Instead, in the case of the enhanced product, the values of the range tend to decrease up to July and increase again in August. We found the same temporal variability in case of $MAR_{1m}$, while in case of $MAR_{5cm}$ we found that the range increases till July and decreases in August. Moreover, a proper fitting of the experimental variograms is achieved for all cases for the enhanced resolution PMW and the MAR derived melt duration. This suggests that the 25 km spatial

resolution might be too coarse to capture the spatial autocorrelation of melting processes. In terms of nugget effect, we found larger values from the MAR outputs than in case of PMW. The decrease in the spatial autocorrelation length in the case of the enhanced product may be a consequence of the local processes that drive melting as the melting season progresses (e.g., impact of bare ice exposure, cryoconite holes, new snowfall, etc.) and of a more developed network of surface meltwater, the presence of supraglacial lakes and, in general, the fact that the processes driving surface meltwater distribution (e.g., albedo,



temperature) promote a stronger spatial dependency of meltwater production at smaller spatial scales. This is even more important when considering that the width of regions such as the bare ice area (where substantial melting occurs) is of the same order of magnitude of the resolution of the coarse PMW dataset. In August, the start of freezing of the surface runoff system and the covering of bare ice, cryoconite holes, together with the draining of the supraglacial lakes and rivers might justify the increase in the range values computed for this month. Our preliminary results, therefore, point to an increased

information content of the enhanced spatial PMW product with respect to the historical, coarse one, offering the opportunity to better capture the spatial details of how surface melting evolved over the Greenland ice sheet over the past ~ 40 years. Further analysis will help to shed light on the processes responsible for the recent acceleration of surface melting.

**5 Conclusions and future work**

       We applied threshold-based melt detection algorithms to the 3.125 km resolution 37 GHz horizontal polarization

passive microwave brightness temperatures to assess the skills of the enhanced PMW product to detect surface melting over the 1979-2019 period over the Greenland Ice Sheet. As the product is composed of data acquired by different sensors onboard of different platforms, we first developed a cross calibration among all the sensors. Then, we compared surface melting detected from PMW data with that estimated from AWS surface/air temperature data and the outputs of the regional climate model MAR. We found that the algorithm making use of a fixed threshold value on $T_b$ values (245 K) and the one based on

the outputs of an electromagnetic model were the most suitable to detect persistent (245K) and sporadic (MEMLS) melting. Overall, we found that that MEMLS algorithm showed the best performance (lowest commission and omission errors). We compared surface melting detected from PMW data with the one estimated by the MAR model when considering the two cases of integrating LWC over the top 5cm and 1m, respectively. We selected these two depths to study those conditions when melting occurs sporadically (5 cm) or persistently (1 m). Following Fettweis et al., (2007), we set up to 0.2 % the minimum

value for the LWC for a pixel in MAR to be flagged as "melting". We obtained good matching (i.e., NSE>0.4 or, at least, positive) in most of the years from 1979-2019 when comparing MEMLS derived melt extent with MAR liquid water content in the first 5 cm of snowpack. In the case of melt extent retrieved by 245K, we found a strong underestimation of melt extent (largelyy negative values of NSE) from 1979 to 1987 likely because of the lower values of "wet" brightness temperature in case of SMMR data, slightly improving from 1993 to 2019 but still negative. Accordingly, the results obtained applying

MEMLS approach are more reliable than in case of 245K algorithm when considering the period 1979 – 2019.

       After assessing the outputs of the PMW-based algorithms, we studied the melt onset date, melt end date, the duration of the melting season and the melting areal extent for the period 1979 – 2019. According to MEMLS algorithm, we found that the melting season has begun 0.404 (0.254) days earlier every year between 1979-2019 (1988-2019) and has ended 0.708 (0.396) days later every year between 1979-2019 (1988-2019). These values are averaged over the whole ice sheet and the

trends are statistically significant at a 95 % level (p-value<0.05). The mean melt duration (MMD) has increased every year by 0.451 d y$^{-1}$ (0.291 d y$^{-1}$) during the period 1979-2019 (1988-2019). We found larger positive and smaller negative trends in



case of 3.125 km product than in case of the 25 km one, possibly because of the higher level of detailing in the coastal regions. When we performed a spatial analysis of the trends for the melt onset dates and duration, we found that the areas where the number of melting days has been increasing are mostly located in West Greenland. The maximum melting surface presents

positive trends as well, with an increment of 0.69% (0.36%) every year respect to the GrIS surface since 1979 (1988).

Finally, we explored the information content of the enhanced resolution dataset with respect to the one at 25 km and the MAR outputs through a semi-variogram approach. The results obtained showed a better fitting of the modelled spherical function to the empirical semi-variogram in case of the 3.125 km and MAR melt duration maps. Our analysis suggests that the enhanced resolution product is sensitive to local scale processes, with higher sensitivity in case of MEMLS algorithm. This

offers the opportunity to improve our understanding of the spatial scale of the processes driving melting and potentially paves the way for using this dataset in statistically downscaling model outputs. In this regard, as a future work, we plan to extend the analysis of spatial scales to the atmospheric drivers of surface melting, such as incoming solar radiation, surface temperature and longwave radiation and complement this analysis with our previous work focusing on understanding the changes in atmospheric patterns that have been promoting enhanced melting in Greenland over the recent decades (Tedesco and Fettweis,

2020). Assessed the capability of this dataset and method in observing temporal trends, a further development can include a combination of the enhanced PMW product with higher resolution satellite data (optical sensors or lower frequencies) in order to investigate the evolution of the surface meltwater networks and the application of  similar tools to other regions, such as the Canadian Arctic Archipelago, the Himalayan Plateau and the Antarctic Peninsula, where the enhancement in spatial resolution can be fully exploited.

**Data availability**

The enhanced resolution passive microwave data are available on the National Snow & Ice Data Center (NSIDC) website at https://nsidc.org/data/NSIDC-0630/versions/1. The meteorological data from Automatic Weather Station can be requested on GC-Net website at http://cires1.colorado.edu/steffen/gcnet/order/admin/station.php. The MAR v3.11.2 outputs are available at ftp://ftp.climato.be/fettweis/MARv3.11/Greenland/ERA_1979-2019-6km/.

**Author contributions**

PC, MT and RR designed the study. PC processed the data and wrote the manuscript. XF processed the MAR model and provided the outputs. All the authors discussed the results and contributed to the paper.

**Competing interests**

The authors declare that they have no conflict of interest.



**Acknowledgements**

Funding for this paper was provided by the US National Science Foundation (Grants ANS 1713072, PLR- 1603331, 1604058, OPP19-01603), by National Aeronautics and Space Administration (grants nos. 80NSSC18K0814, 80NSSC17K0351 and NNX17AH04G), by the Heising-Simons Foundation (HS 2019 – 1160) and by AICS-Italian Agency for Development Cooperation (D24I20000430005). The first author also thanks for the support provided by the University of Brescia and the
H2CU-Honors Center of Italian Universities.

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





**Figure 1: Maps of passive microwave brightness temperature at 37 GHz, horizontal polarization, acquired over Greenland on 16 July 2001 in the case of the a) coarse and b) enhanced products. Panels c) and d) refer to the area highlighted in the square in panels a) and b).**



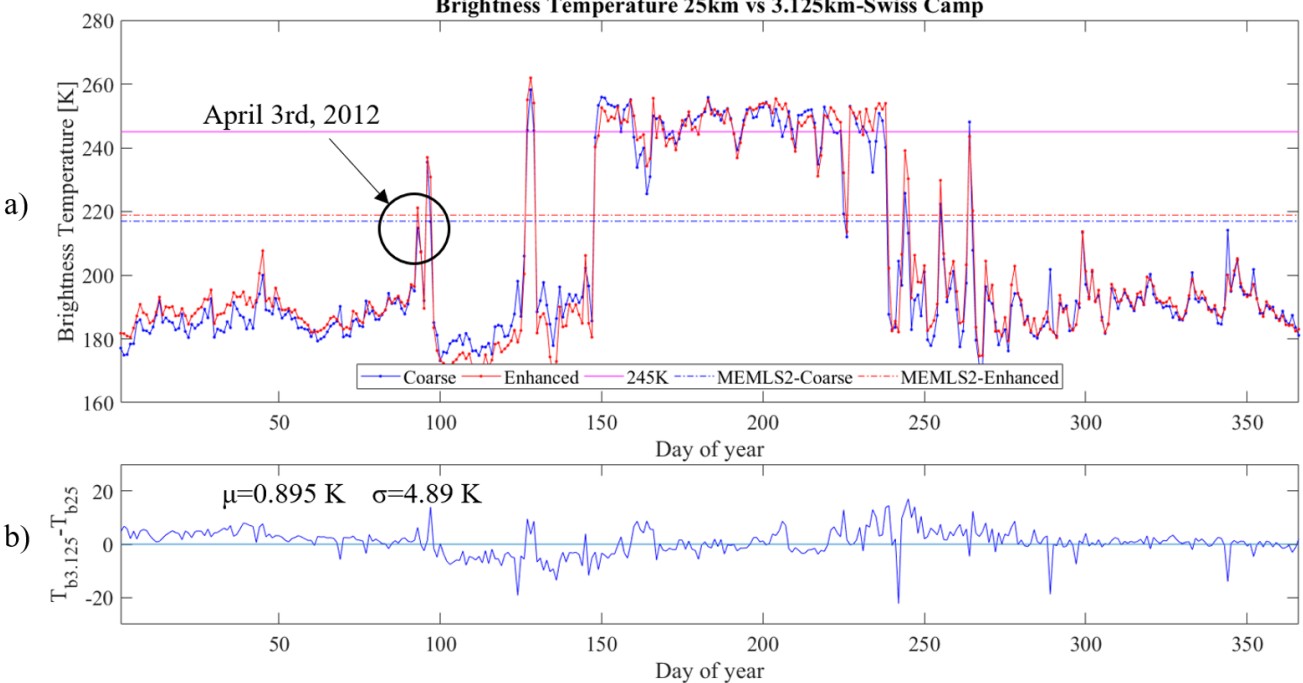

**Figure 2 a) Time series of brightness temperatures at 37 GHz, horizontal polarization, for the year 2012 of the pixel containing Swiss Camp site (69° 34' 06" N, 49° 18' 57" W, ) in the case of the coarse (blue) and enhanced (red) products. Threshold values, shown as horizontal lines, are obtained from two approaches considered in this study: 245K and MEMLS. b) Difference between the 3.125 km and the 25 km $T_b$ time series for the same pixel.**

735



| Station | Latitude | Longitude | Elevation [m a.s.l.] |
|---|---|---|---|
| Swiss Camp | 69° 34' 06" N | 49° 18' 57" W | 1149 |
| Crowford Pt. 1 | 69° 52' 47" N | 46° 59' 12" W | 2022 |
| NASA-U | 73° 50' 31" N | 49° 29' 54" W | 2369 |
| GITS | 77° 08' 16" N, | 61° 02' 28" W | 1887 |
| Humboldt | 78° 31' 36" N | 56° 49' 50" W | 1995 |
| Summit | 72° 34' 47" N | 38° 30' 16" W | 3254 |
| TUNU-N | 78° 01' 0" N | 33° 59' 38" W | 2113 |
| DYE-2 | 66° 28' 48" N | 46° 16' 44" W | 2165 |
| JAR-1 | 69° 29' 54" N | 49° 40' 54" W | 962 |
| Saddle | 66° 00' 02" N | 44° 30' 05" W | 2559 |
| South Dome | 63° 08' 56" N | 44° 49' 00" W | 2922 |
| NASA-E | 75° 00' 00" N | 29° 59' 59" W | 2631 |
| Crowford Pt. 2 | 69° 54' 48" N | 46° 51' 17" W | 1990 |
| NASA-SE | 66° 28' 47" N | 42°30' 00" W | 2425 |
| KAR | 69° 41' 58" N | 33° 00' 21" W | 2579 |
| JAR-2 | 69º 25' 12" N | 50º 03' 27" W | 568 |
| KULU | 65° 45' 30" N | 39° 36' 06" W | 878 |

**Table 1 Locations of the automatic weather stations of the selected Greenland Climate Network (GC-Net).**

740

|  | SMMR | SSM/I (F08) | SSM/I (F11) | SSM/I (F13) | SSMI/S (F17) |
|---|---|---|---|---|---|
| **Platform** | NIMBUS-7 | DMSP-F08 | DMSP-F11 | DMSP-F13 | DMSP-F17 |
| **Temporal coverage** | 26/10/1978-20/08/1987 | 09/07/1987-30/12/1991 | 03/12/1991-30/09/1995 | 03/05/1995-01/04/2009 | 04/11/2006-operating |
| **Frequency (GHz)** | 37 | 37 | 37 | 37 | 37 |
| **IFOV (Ka band) [km$^2$]** | 27*18 | 37*28 | 37*28 | 37*28 | 37*28 |
| **Incidence Angle** | 50.2° | 53.1° | 53.1° | 53.1° | 53.1° |
| **Swath width [km]** | 780 | 1400 | 1400 | 1400 | 1700 |
| **Data acquisition** | Alternate days | Daily | Daily | Daily | Daily |
| **Ascending Equator Crossing Time** | 24:00 | 18:17 | 18:25 | 17:43 | 18:33 |
| **Descending Equator Crossing Time** | 12:00 | 06:10 | 05:00 | 05:51 | 07:08 |

**Table 2: Characteristics of the selected passive microwave sensors.**



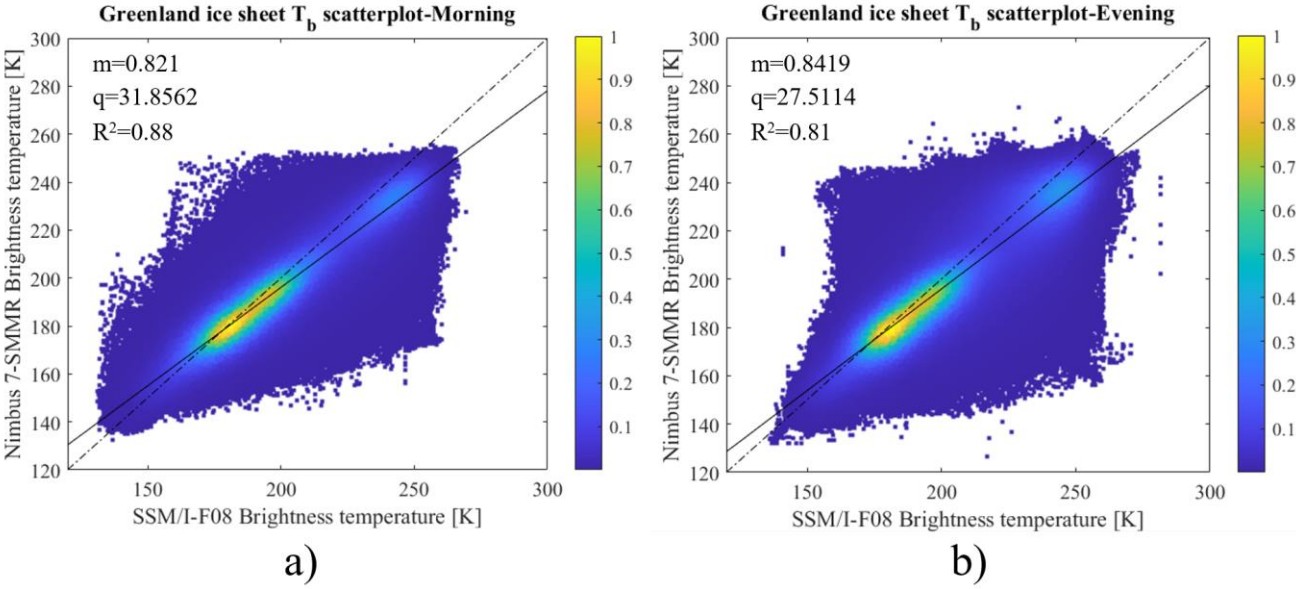

**Figure 3 Density scatter plots of SMMR and SSM/I-F08 brightness temperature data sensed during the overlap period (09/07/1987-20/08/1987) of the two sensors over the Greenland ice sheet for a) morning and b) evening passes. Solid black lines show the linear fitting and dashed black lines the 1:1 line. The colour palette indicates points' density.**



| Platforms | SMMR-F08 | | F08-F11 Evening | | F11-F13 Evening | | F13-F17 Evening | |
|---|---|---|---|---|---|---|---|---|
| | **Evening** | **Morning** | **Evening** | **Morning** | **Evening** | **Morning** | **Evening** | **Morning** |
| NSE | 0.898 | 0.936 | 0.999 | 0.999 | 0.999 | 0.999 | 0.997 | 0.997 |
| Average difference [K] | -4.27 | -3.43 | 0.50 | 0.24 | -0.49 | 0.02 | 0.17 | 0.52 |

**Table 3 Average enhanced resolution $T_b$ differences at 37 GHz, horizontal polarization for the different PMW sensors and NSE coefficients computed for the histograms of brightness temperatures.**

750




| Greenland SMMR vs. SSM/I-F08 | | | | | | | |
|---|---|---|---|---|---|---|---|
| **X=F08** | $m_1$ | $m_2$ | $q_1$ | $q_2$ | $R^2_2$ | $d_1$ | $d_2$ |
| **Morning** | 0.818 | 0.821 | 32.387 | 31.856 | 0.88 | 0.69 | 0.46 |
| **Evening** | 0.849 | 0.842 | 26.027 | 27.511 | 0.81 | 0.12 | 0.33 |
| **X=SMMR** | $m_1$ | $m_2$ | $q_1$ | $q_2$ | $R^2_2$ | $d_1$ | $d_2$ |
| **Morning** | 1.075 | 1.0722 | -11.140 | -10.581 | 0.88 | 0.56 | 0.52 |
| **Evening** | 0.964 | 0.9653 | 11.424 | 11.123 | 0.81 | 0.09 | 0.12 |
| **Greenland F08 vs. F11** | | | | | | | |
| **X=F11** | $m_1$ | $m_2$ | $q_1$ | $q_2$ | $R^2_2$ | $d_1$ | $d_2$ |
| **Morning** | 0.991 | 0.989 | 2.041 | 2.537 | 0.98 | 0.31 | -2.91 |
| **Evening** | 0.998 | 1.002 | 0.979 | -0.010 | 0.98 | 0.25 | 0.31 |
| **X=F08** | $m_1$ | $m_2$ | $q_1$ | $q_2$ | $R^2_2$ | $d_1$ | $d_2$ |
| **Morning** | 0.987 | 0.995 | 2.230 | 0.528 | 0.98 | 0.10 | 0.26 |
| **Evening** | 0.980 | 0.980 | 3.332 | 3.711 | 0.98 | 0.08 | 0.11 |
| **Greenland F11 vs. F13** | | | | | | | |
| **X=F13** | $m_1$ | $m_2$ | $q_1$ | $q_2$ | $R^2_2$ | $d_1$ | $d_2$ |
| **Morning** | 0.996 | 1.001 | 2.328 | -0.262 | 0.98 | 0.11 | 0.14 |
| **Evening** | 0.981 | 0.985 | 3.831 | 0.185 | 0.99 | -4.82 | -4.98 |
| **X=F11** | $m_1$ | $m_2$ | $q_1$ | $q_2$ | $R^2_2$ | $d_1$ | $d_2$ |
| **Morning** | 0.962 | 0.977 | 8.322 | 4.482 | 0.98 | -1.73 | -0.32 |
| **Evening** | 0.998 | 1.002 | 0.934 | 0.185 | 0.99 | 0.10 | 0.28 |
| **Greenland F13 vs. F17** | | | | | | | |
| **X=F17** | $m_1$ | $m_2$ | $q_1$ | $q_2$ | $R^2_2$ | $d_1$ | $d_2$ |
| **Morning** | 1.019 | 1.029 | -3.029 | -5.013 | 0.98 | -0.11 | -0.005 |
| **Evening** | 1.004 | 1.007 | -0.438 | -1.161 | 0.99 | 0.14 | 0.20 |
| **X=F13** | $m_1$ | $m_2$ | $q_1$ | $q_2$ | $R^2_2$ | $d_1$ | $d_2$ |
| **Morning** | 0.959 | 0.953 | 7.267 | 8.370 | 0.98 | -0.19 | -0.35 |
| **Evening** | 0.982 | 0.982 | 3.200 | 3.205 | 0.99 | 0.27 | 0.25 |

**Table 4: Slope (m) and intercept (q) obtained from the linear regression analysis between SMMR and SSM/I-F08 enhanced PMW brightness temperatures at 37 GHz, horizontal polarization over Greenland. The subscripts refer to the case when the coefficients are weighted by means of the $R^2$ (case 1, see Eq. (5) and Eq. (6)) or not (case 2). In the Table, we also report the values for the $R^2$ as well as the values of d computed according to Eq. (9).**





**Figure 4: Brightness temperature histograms before and after the application of the intercalibration relations, Greenland. Relations are applied for both evening (a) and morning (b) passes, reporting the histograms of the data and the distance between the histograms for original data. The first column represents the uncorrected data, the second the results applying the correction to SMMR data and the third the results applying the correction to the SSM/I data.**

760



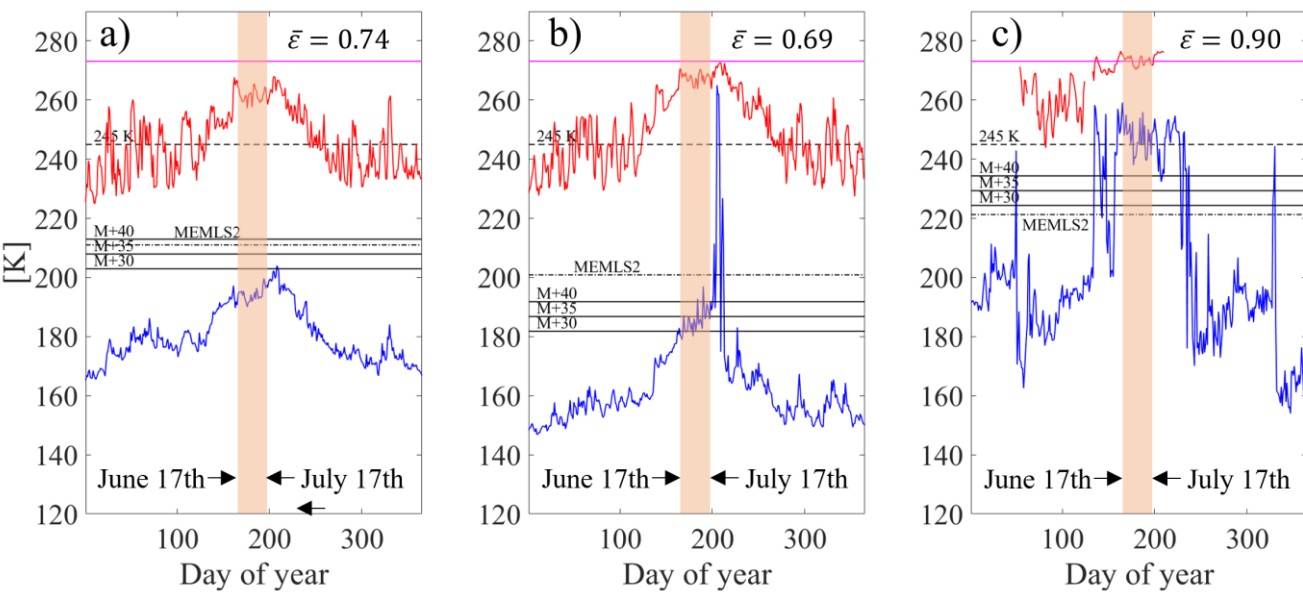

**Figure 5: Timeseries of enhanced resolution brightness temperature 37 GHz H-pol (blue) and air temperature (red) at Summit (a),
Humboldt (b) and Swiss Camp (c) stations, year 2005. Threshold values obtained with the different detection algorithms are also
reported as horizontal black lines (solid M+DT, dashed 245K and dash-dot MEMLS) while the 0°C threshold is reported as magenta
solid line. The 30 days window between 17 June and 17 July is shown in the shaded orange area, reporting the values of average
estimated emissivity (ε).**





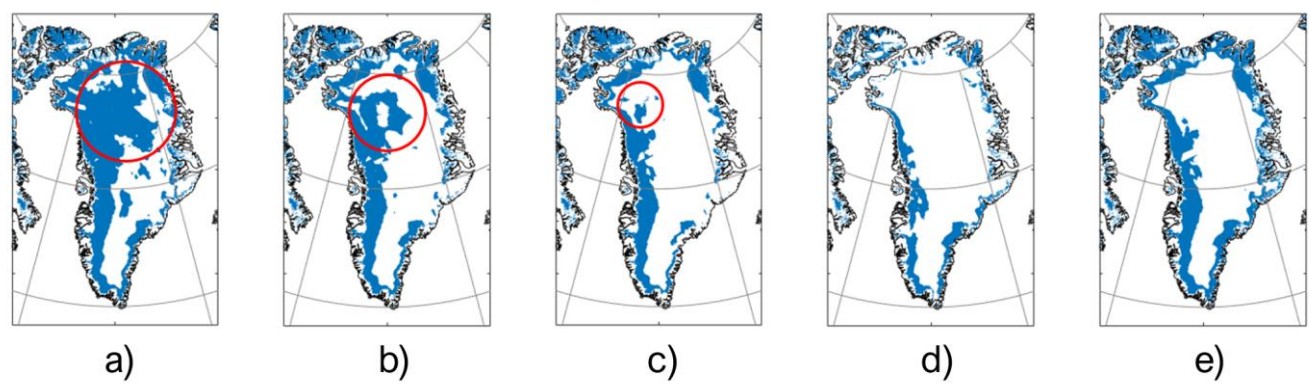

770

**Figure 6: Melting maps obtained applying a) M+30, b) M+35, c) M+40, d) 245K and e) MEMLS algorithms over Greenland Ice sheet on July 13th, 2008. Example of areas presenting the false detection problem are within the red circle.**





| | | Average % Commission | Average % Omission | C+O % | C+O Mean % |
|---|---|---|---|---|---|
| M+30 | T=0°C | 3.71 | 2.63 | 6.34 | 7.79 |
| | T=-1°C | 3.04 | 3.78 | 6.82 | |
| | T=-2°C | 2.44 | 5.66 | 8.1 | |
| | LWC1m | 7.11 | 1.51 | 8.62 | |
| | LWC5cm | 7.01 | 2.07 | 9.07 | |
| M+35 | T=0°C | 2.34 | 3.19 | 5.53 | 6.83 |
| | T=-1°C | 1.83 | 4.5 | 6.33 | |
| | T=-2°C | 1.37 | 6.35 | 7.72 | |
| | LWC1m | 5.5 | 1.83 | 7.33 | |
| | LWC5cm | 4.78 | 2.48 | 7.26 | |
| M+40 | T=0°C | 1.73 | 3.98 | 5.72 | 6.84 |
| | T=-1°C | 1.3 | 5.37 | 6.68 | |
| | T=-2°C | 0.93 | 7.32 | 8.25 | |
| | LWC1m | 4.49 | 2.23 | 6.72 | |
| | LWC5cm | 3.88 | 2.98 | 6.87 | |
| MEMLS | T=0°C | 2.7 | 2.38 | 5.08 | 6.66 |
| | T=-1°C | 2.13 | 3.62 | 5.76 | |
| | T=-2°C | 1.63 | 5.44 | 7.07 | |
| | LWC1m | 6.33 | 1.49 | 7.81 | |
| | LWC5cm | 5.52 | 2.04 | 7.56 | |
| 245K | T=0°C | 0.63 | 5.38 | 6.01 | 6.92 |
| | T=-1°C | 0.46 | 7.02 | 7.48 | |
| | T=-2°C | 0.31 | 9.19 | 9.51 | |
| | LWC1m | 2.58 | 2.95 | 5.53 | |
| | LWC5cm | 2.23 | 3.83 | 6.06 | |

775

**Table 5: Commission and omission errors (as percentage of the total number of days considered) averaged over all the selected automatic weather stations in Greenland.**



**Figure 7: LWC from MAR averaged in the first 5 cm (a) and first 1 m (b) of the snowpack, (c) time series of 37 GHz horizontal polarization brightness temperature (3.125 km), surface air temperature from AWS and 245 K (dashed purple line), M+ΔT (solid purple lines) and MEMLS (dash-dotted purple line) thresholds for Swiss Camp site in 2001.**



**Figure 8: Melt extent estimation from PMW 37GHz H-pol brightness temperature and the regional climate model MAR. Timeseries were obtained using the 245K algorithm and LWC average in the first 1 m of snowpack (left) and the MEMLS algorithm and LWC average in the first 5 cm of snowpack (right), for the years (a) 1984 and (b) 2006.**



| Year | NSE 245K/MAR$_{1m}$ | NSE MEMLS/MAR$_{5cm}$ | Year | NSE 245K/MAR$_{1m}$ | NSE MEMLS/MAR$_{5cm}$ |
|---|---|---|---|---|---|
| 1979 | -128.769 | -0.79225 | 2000 | -5.57807 | 0.879176 |
| 1980 | -278.146 | -2.91691 | 2001 | -10.9472 | 0.770768 |
| 1981 | -173.495 | -0.88112 | 2002 | -6.55333 | 0.730885 |
| 1982 | -176.464 | -1.25147 | 2003 | -13.279 | 0.727126 |
| 1983 | -151.596 | -0.54018 | 2004 | -7.82663 | 0.681861 |
| 1984 | -144.117 | -1.6115 | 2005 | -5.36981 | 0.78231 |
| 1985 | -267.337 | -2.63938 | 2006 | -5.24982 | 0.74654 |
| 1986 | -128.639 | -1.57304 | 2007 | -4.85843 | 0.823594 |
| 1987 | -39.5243 | -1.89319 | 2008 | -9.04729 | 0.700948 |
| 1988 | -35.1243 | -0.29856 | 2009 | -5.2188 | 0.770134 |
| 1989 | -22.7824 | -0.02961 | 2010 | -8.35154 | 0.63791 |
| 1990 | -41.5149 | -0.34192 | 2011 | -4.59082 | 0.882278 |
| 1991 | -31.6141 | -0.42211 | 2012 | -3.40038 | 0.851023 |
| 1992 | -10.9036 | 0.892694 | 2013 | -8.61788 | 0.760056 |
| 1993 | -6.45558 | 0.818286 | 2014 | -9.78458 | 0.646061 |
| 1994 | -11.2671 | 0.52946 | 2015 | -11.4177 | 0.611012 |
| 1995 | -7.76444 | 0.020502 | 2016 | -11.8271 | 0.505357 |
| 1996 | -10.2121 | 0.511991 | 2017 | -90.9062 | -0.3225 |
| 1997 | -6.44938 | 0.770647 | 2018 | -35.901 | 0.484603 |
| 1998 | -8.26258 | 0.605346 | 2019 | -39.9834 | 0.319033 |
| 1999 | -4.20066 | 0.86495 | | | |

**Table 6: Nash-Sutcliffe Efficiency coefficients computed for the comparison of retrieved melt extent using 245K (MEMLS) algorithms applied to the enhanced resolution PMW brightness temperatures and MAR liquid water content outputs averaged in the first 1 m (5 cm) of the snowpack.**



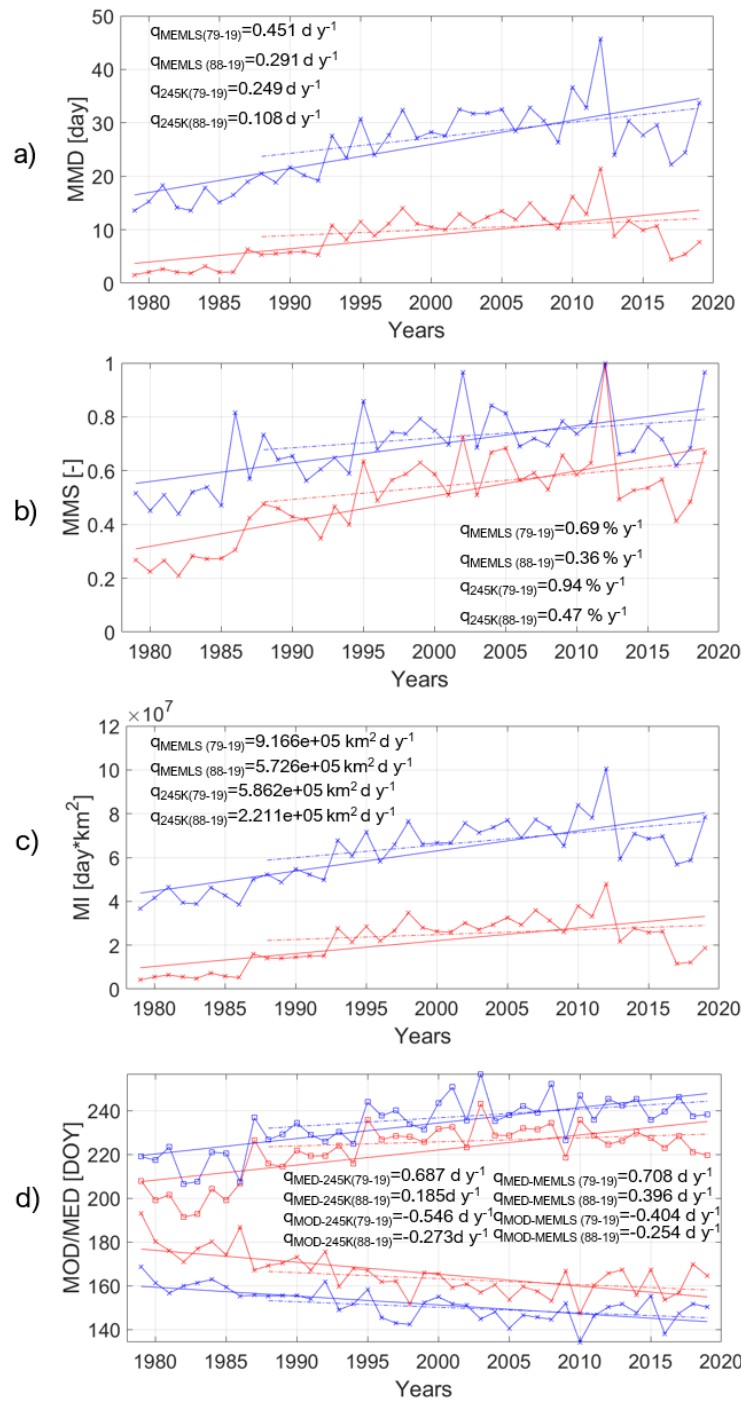

**Figure 9** Time series of annual a) mean melt duration, b) maximum melting surface fraction, c) melt index and d) melt onset date and melt end date. Regression lines computed for the periods 1979-2019 (solid line) and 1988-2019 (dashed-dot line). MMD is averaged over all the GrIS pixels. Red (blue) lines refer to 245K (MEMLS) algorithm while in panel d) squares (crosses) refer to MED (MOD).







**Figure 10 Maps of 95%-significant trends (1979-2019) obtained with 245K (a, c, e) and MEMLS (b,d,f) algorithms for melt duration (MD panels a and b), melt onset date (MOD, panels c and d) and melt end date (MED, panels e and f). MOD and MED are defined as the first and last two melting days in a row.**





**Figure 11: Empirical (blue crosses) and modelled (red line) semi-variograms for Greenland melt duration computed applying the MEMLS, panels a) and b), and 245K, panels c) and d), to both 25 km, a) and c), and 3.125 km, b) and d), resolution data for each month of the melting season (May, June, July and August). The range (r), sill (s), nugget (n) and R² values are reported.**





| | May | | June | | July | | August | |
|---|---|---|---|---|---|---|---|---|
| | $MAR_{1m}$ | $MAR_{5cm}$ | $MAR_{1m}$ | $MAR_{5cm}$ | $MAR_{1m}$ | $MAR_{5cm}$ | $MAR_{1m}$ | $MAR_{5cm}$ |
| r | 199.17 | 187.70 | 233.05 | 207.26 | 186.16 | 282.57 | 211.7 | 230.32 |
| s | 3.97 | 4.02 | 18.28 | 17.78 | 19.66 | 5.24 | 14.08 | 1.78 |
| n | 3.35 | 4.66 | 44.97 | 37.94 | 79.79 | 31.86 | 28.64 | 5.58 |
| $R^2$ | 0.2 | 0.34 | 0.41 | 0.48 | 0.24 | 0.14 | 0.38 | 0.12 |

**Table 7: Parameters of the spherical function fitted to the empirical semi-variogram for the maps of melt duration obtained cumulating the LWC simulated by MAR over the first 1m and 5 cm of snowpack.**