# Peer review of "Surface melting over the Greenland ice sheet derived from enhanced resolution passive microwave brightness temperatures (1979 – 2019)"

_The Cryosphere, 2020_

## Referee Comment (RC1) · Anonymous Referee #1 · 25 Nov 2020

General comments:

This study analyzes the newly developed NASA MEaSUREs calibrated enhanced resolution (∼ 3.125 km) passive microwave dataset (37 GHz horizontally polarized channel) (Brodzik et al., 2016, cited in this paper) to examine whether the dataset can be used for studies on the Greenland ice sheet (GrIS) surface melt. The dataset was developed by using the data from the following satellite microwave radiometers: The Nimbus-7 Scanning Multichannel Microwave Radiometer (SMMR) and Defense Meteorological Satellite Program (DMSP) Special Sensor Microwave/Imager (SSM/I). Because the frequency of the GrIS surface melt has been increasing recently due to the ongoing

rapid warming, the GrIS surface melt commands considerable attention. Therefore, the topic explored by the authors fits very well with the scope of this journal.

In this paper, the authors compare five post-processing methods applied to the new dataset: the M+ΔT methods with changing ΔT values of 30, 35, and 40 K, the 245 K fixed threshold method, and the MEMLS (Microwave Emission Model of Layered Snowpack) method. All these five methods can be categorized into the threshold-based method. The first four methods are very simple, whereas the MEMLS method is relatively physically based but its threshold value does not change dynamically. These methods give threshold values of passive microwave brightness temperatures to detect the surface melt. In case a (measured and) post-processed value from a satellite becomes higher than a threshold value, the occurrence of the surface melt can be estimated. Based on the comparisons of the melt detection results with in-situ meteorological/snow data from automated weather stations on the GrIS and the regional climate model MAR, the authors conclude that the MEMLS method shows the best performance in terms of capturing the GrIS surface melt. Finally, the authors present inter-annual variations of the GrIS surface melt area extent obtained from this study.

My honest impression is that this paper contains so many information that many readers will find it difficult to follow the discussion. Tedesco et al. (2013, cited in this paper) already have demonstrated the effectiveness of the MEMLS method over the GrIS, so that, I think results from the M+ΔT methods and the constant 245 K method can be removed. It is because they are very simple compared to the MEMLS method. I do not find interests showing these results in this paper. The authors also compare their results with the outputs from the MAR model. I completely agree with the point that the MAR model is very sophisticated; however, the model output is not the reality. Therefore, I cannot understand why the authors want to compare them in this study, although I have confirmed from Figure 7 again that the MAR model performs very well over the GrIS.

In the global scientific community studying the GrIS surface melt, the dataset by Mote

(2007), which utilizes data from the 18 and 19.35 GHz horizontally polarized channels in the same sensors/satellites as those used in this study, has long been utilized widely. As fat as I know, the dataset employs a dynamically changing threshold method to detect the GrIS surface melt. Because the horizontal resolution of the dataset by Mote (2007) is 25 km, it seems to me that the new dataset has a big advantage. Therefore, the authors should compare their MEMLS-method-based results with the dataset by Mote (2007). Without this, readers cannot know advantages/significances of the new dataset presented in this study.

Also, I would like to suggest that the data and methods section (Sect. 2) is a mix-up of data, methods, results, and discussion, which confuses readers. Figures 2, 3, and 4, as well as Tables 3 and 4 should be presented in the results and discussion section. Please reformulate the section.

I would like to suggest that the authors should attend to the above-mentioned major issues before considering its publication.

Other specific comments are as follows:

Specific comments:

L. 25: More detailed explanation of "local scale processes" is needed here.

L. 86 $\sim$ 90: It is necessary to introduce why such a high-resolution dataset from the Ka band product were not available until recently. What is the key innovation that enabled us to use the Ka band data for the detection of the ice sheet surface melt? It is also important to explain the difference in sensitivities of the K and Ka bands data to the liquid water clouds.

L. 130: "2.2 Greenland air/surface temperature data": It is necessary to explain how the authors obtain surface temperature from the AWS (automated weather station) data. It is because the AWSs do not measure surface temperature directly.

L. 132 $\sim$ 133: Strictly speaking, even if the surface temperature reaches 0 degreeC, it

does not ensure that meltwater exits at the surface. How do the authors detect whether meltwater exits at the surface or not from the AWS data?

L. 185 ∼ 186: "Building on Tedesco (2009), we considered the two LWC values of 0.1 % and 0.2 %": Please explain more in detail about this process. It is unclear why 0.1 and 0.2% are chosen here.

L. 193: For MEMLS, why do the authors consider only the case of 0.2% LWC?

L. 193 ∼ 196: "As we explain below, this choice was driven by the performance of the different considered algorithms. Moreover, we found that the fixed-threshold algorithm is more sensitive to persistent melting where the MEMLS-based one can detect sporadic melting. This allows us to analyze both melting conditions (sporadic vs. persistent) and analyze them within the long-term, large spatial scales that the PMW data can provide.": I think it is not necessary to state them here. They can be removed.

L. 304: "with the MEMLS being the most sensitive": The authors' intention is unclear. Sensitive to what?

Technical corrections:

L. 16: "MeASUREs": Its definition should be indicated here.

L. 17: "Km" -> "km"

L. 19: "MEMLS model": Brief explanation of the model or the definition of the abbreviation should be indicated here.

L. 82: Please provide the definition of the abbreviation "rSIR".

L. 103: "SMMR": Please provide its definition here.

L. 131: "In order, to" -> "In order to"

L. 146 ∼ 147: "Lateral and lower boundary conditions are prescribed from reanalysis datasets." -> "Lateral and lower boundary conditions of the atmosphere are prescribed

from reanalysis datasets."

L. 153: "meltwater extent" -> "melt extent"

L. 195: "where" -> "whereas"

L. 211 $\sim$ 214: Please follow the instruction how to indicate date and time in the text. https://www.the-cryosphere.net/submission.html#math

L. 576 $\sim$ 579: Brodzik et al. is updated in 2020.

References:

Mote, T. L.: Greenland surface melt trends 1973–2007: evidence of a large increase in 2007, Geophys. Res. Lett., 34, L22507, https://doi.org/10.1029/2007GL031976, 2007.

---

## Referee Comment (RC2) · Anonymous Referee #2 · 30 Nov 2020

General Comments

The authors present analysis using a new, higher resolution passive microwave dataset for determining surface melt across the Greenland Ice Sheet. The authors make a strong case for why such a dataset is important for monitoring the ice sheet and demonstrate that the higher resolution data allows us to study surface melt in greater detail, altering the magnitude of some of the temporal trends and providing sufficient resolution for more thorough spatial analyses. The work is novel, presenting a new dataset and analyzing it with an existing algorithm to study trends in surface melt extent and timing.

[Figure]

The methods implemented are appropriate and sufficiently explained in most cases. In my specific comments, I have a few points that I would like to see addressed in terms of articulating implications of some of the issues the authors note with the data (i.e. differences in the four PMW sensors used, issue of poor matching between MEMLS and MAR5cm before 1992, MEMLS algorithm issues after main melt season). I do not consider any of these to be major issues; I would just like to see some clarification and explanation of the potential effects of this issues on the results. Additionally, an overall comparison of how this PMW melt detection compares to other PMW melt products in terms of commission and omission errors should be included in order to put this work into context.

The results are significant, demonstrating that trends in surface melting are sensitive to the scale at which they are studied. The trends identified are important in our assessment of surface processes that affect mass balance and sea level rise. The surface melt product is an important dataset that can be used in future work as described in the conclusions.

The manuscript is overall well-written and flows logically, with only minor issues that will be easy to fix.

Please find my specific comments and technical corrections below.

Specific Comments

line 171: Please explain what sigma is. It "varies in space and time" based on what – is it the standard deviation?

line 190: I appreciate that many melt threshold/algorithms are implemented (and that they are compared to both in-situ data and the MAR output). Please explain why you selected the threshold/algorithms as you did given that you also presented at least 2 others.

line 247: With respect to the differences in acquisition time, is there a consistent

lead/lag between timing of SMMR and SSM/I-F08? If so, how might the directional-ity of the lead/lag impact the analysis?

line 260: Which correction did you apply to the SMMR data (the first method with weighted values or the second method using all values and the least square fitting) and why?

line 260: What are the implications (if any) of not correcting the datasets? For instance, the average differences from F08-F11 and F13-F17 are positive, while the difference for F11-F13 is negative in the evening and positive (close to zero) in the morning. If agreement is worse when corrected, I agree that it makes sense not to implement the linear corrections, but it would be important to address what the potential effect of this is.

line 306: Is there an emissivity threshold being considered here to indicate if melting is or is not occurring? If so, please add that.

line 307: I think it was meant to say lower than in Summit Camp case?

line 315: (AWS Comparison Section) Were there any temporal trends the commis-sion/omission errors of the melt algorithms as compared to the AWS data?

lines 325-327: I think the numbers for LWC1m and LWC5cm were swapped here?

lines 346-346: You bring up a very interesting point here. Because the brightness temperature after the largest part of the melt season has ended up lower than the Jan/Feb average, then the MEMLS algorithm would be less able to detect subsequent melt events. Is this a consistent pattern that is observed across sites/years? This could lead to a change in the frequency of omission errors of the MEMLS algorithm pre and post main melt season. Please discuss potential implications of this issue.

line 355: How do the commission and omission errors for these algorithms compare to other PMW melt detection products.

lines 369-371: Please explain your decision to compare 245K and MAR1m and also compare MEMLS and MAR5cm. I believe it is because the expected differences in sensitivity of each of the different methods of detecting melt, but just want to be sure that is why this decision was made.

lines 375-380: It seems that using the SMMR data (from 1979-1987) is part of the issue here. Is that correct? Is it partly because of the difference in time of day? Or difference in sensor technology used?

line 382: What are the implications of the melt extent being underestimated?

lines 396-397: Is there precedent for using this definition of MOD and MED?

line 406: Here and elsewhere you refer to the trends in the coarser-resolution data. Please consider including this analysis in supplemental material.

line 436: Is there any explanation for the areas in the map with anomalous trends? (figure f, negative trend in Northern Greenland, figure d, positive trend in some regions in central Greenland)

lines 436-438: How are pixels that do not consistently (every year) experience melt handled?

lines 440-446: This content reads more like methods. Consider relocating the description of the methods of the semi-variogram analysis.

line 452-453: The comment about extending this analysis seems out of place.

line 457: Consider showing the figures that accompany the data for Table 7. Perhaps in the supplement at least?

line 458: These are semi-variograms for melt duration in each month. Is this the number of days of the month that melt occurs for a given pixel? Do the days need to be consecutive? I think more detail about the melt duration variable should be provided here.

lines 464-468: Is there a way to compute uncertainty associated with these distances?

lines 475-476: What does the larger nugget value for MAR as compared to the PMW data tell us?

lines 499-500: The sentence about the threshold for melting seems out of place in the conclusions.

line 500-501: The data do not seem to support "good matching" in most of the years from 1979-2019. The data do seem to support good matching from 1992-2019. Please add this caveat to the statement.

Technical Corrections

line 11: modulation "of" ice dynamics

line 13: "in view of" should perhaps be replaced with "due to"?

line 17: km instead of Km

line 19: capable "of detecting"

line 25: the word "interest" seems out of place. Delete or replace, perhaps with "usefulness"?

line 26: monitor should be "monitoring"

line 74: here and elsewhere you use Tbs, when I think leaving it singular as Tb is more clear

line 188: Should this be Tc or Tb? If it is meant to say TÂňc, please define this term.

line 190: "as sensitivity to Zwally. . ." not sure what is meant here. Typo?

line 251: R2 needs to be a superscript 2

line 255: specify that you referring to data in Table 4 here
line 267: move "daily averaged from AWS" to directly after air/surface temperature to improve sentence clarity

line 281: correct 919% to 9.19%

line 305: fix subscript on Tb

line 336: Is this average surface air temperature? Please specify.

line 342: detected by the threshold algorithms in AWS temperature? By all three?

line 344: should be "corresponds"

line 355: consider describing it not as an overall error but as what it is a mean of errors calculated using different techniques.

line 364: perhaps rephrase as "Here, we remind the reader that.."

lines 399-400: typo? Partial repeating of a line

lines 405-406: typo in years indicated here?

line 418: Fix figure numbers

line 468: some missing words, should read "the value of the range results is lower in the case of.."

line 473: "till" should be "until"

line 503: typo of word largely

line 525: Perhaps add "We have" to "assessed the capability…."

line 729: (Figure 1) consider including scale bar for figures c and d

line 739: (Table 1) table caption perhaps should say "of the selected Greenland Climate Network (GC-Net) sites"

line 752: (Table 4) This table shows regression analysis for more comparisons than

just SMMR and SSM/I-F08. Please update caption to reflect this.

line 760: (Figure 4) Please ensure that y-axis are the same for all three panels of figure 4a

line 771: (Figure 6) Please consider adding labels to each map for ease of interpreting the figure

line 780: (Figure 7) Please indicate what the vertical teal lines represent

---

## Author Comment (AC1) · 24 Jan 2021

Dear reviewer,

Please find below our reply. We thank you for your comment and precious suggestions.

P. Colosio, M.Tedesco, X. Fettweis and R. Ranzi
General comments: This study analyzes the newly developed NASA MEaSUREs calibrated enhanced resolution (âĹij 3.125 km) passive microwave dataset (37 GHz horizontally polarized channel) (Brodzik et al., 2016, cited in this paper) to examine whether the dataset can be used for studies on the Greenland ice sheet (GrIS) surface melt. The dataset was developed by using the data from the following satellite microwave radiometers: The Nimbus-7 Scanning Multichannel Microwave Radiometer (SMMR) and Defense Meteorological Satellite Program (DMSP) Special Sensor Microwave/Imager (SSM/I). Because the frequency of the GrIS surface melt has been increasing recently due to the ongoing rapid warming, the GrIS surface melt commands considerable attention. Therefore, the topic explored by the authors fits very well with the scope of this journal. In this paper, the authors compare five post-processing methods applied to the new dataset: the M+$\Delta$T methods with changing $\Delta$T values of 30, 35, and 40 K, the 245 K fixed threshold method, and the MEMLS (Microwave Emission Model of Layered Snowpack) method. All these five methods can be categorized into the threshold based method. The first four methods are very simple, whereas the MEMLS method is relatively physically based but its threshold value does not change dynamically. These methods give threshold values of passive microwave brightness temperatures to detect the surface melt. In case a (measured and) post-processed value from a satellite becomes higher than a threshold value, the occurrence of the surface melt can be estimated. Based on the comparisons of the melt detection results with in-situ meteorological/snow data from automated weather stations on the GrIS and the regional climate model MAR, the authors conclude that the MEMLS method shows the best performance in terms of capturing the GrIS surface melt. Finally, the authors present inter-annual variations of the GrIS surface melt area extent obtained from this study. My honest impression is that this paper contains so many information that many readers will find it difficult to follow the discussion. Tedesco et al. (2013, cited in this paper) already have demonstrated the effectiveness of the MEMLS method over the GrIS, so that, I think results from the M+$\Delta$T methods and the constant 245 K method can be removed. It is because they are very simple compared to the MEMLS method. I do not

find interests showing these results in this paper.

R: We thank the reviewer for this useful comment and we acknowledge that Tedesco et al. 2013 performed an analysis of the algorithms. Yet, we would like to keep the description of those results for two reasons: 1) we think that adding another paper to the comparison of the different algorithms increases the confidence in our results. It is a best practice in science to test the robustness of the results from previous studies and this is one of the reasons to perform and show the comparison; 2) the results are discussed here with respect to the enhanced product. Given that the previous work was performed on the 25 km and the methods used to create the gridded values are different (not only in terms of spatial resolution but in terms of how Tbs are computed and extrapolated form the observations), we think that it is important to show that the results from the previous study still hold. Again, we thank the reviewer for this useful suggestion.

The authors also compare their results with the outputs from the MAR model. I completely agree with the point that the MAR model is very sophisticated; however, the model output is not the reality. Therefore, I cannot understand why the authors want to compare them in this study, although I have confirmed from Figure 7 again that the MAR model performs very well over the GrIS.

R: In this paper we compare our results with the outputs from the MAR model. In this regard, we do not use the modelled data as a validation but as an assessment, introducing a further comparison to evaluate the different algorithms. In particular, using a third melt classification dataset lets us better understand where the PMW and AWS melt detection techniques are in agreement. Moreover, MAR realistic assimilation and modelling of observed data is available on a gridded geographic support which enables an effective comparison with microwave data.

In the global scientific community studying the GrIS surface melt, the dataset by Mote (2007), which utilizes data from the 18 and 19.35 GHz horizontally polarized channels

in the same sensors/satellites as those used in this study, has long been utilized widely. As far as I know, the dataset employs a dynamically changing threshold method to detect the GrIS surface melt. Because the horizontal resolution of the dataset by Mote (2007) is 25 km, it seems to me that the new dataset has a big advantage. Therefore, the authors should compare their MEMLS-method-based results with the dataset by Mote (2007). Without this, readers cannot know advantages/significances of the new dataset presented in this study.

R: We thank the reviewer for suggesting this further comparison. We agree with the reviewer that a comparison with a coarser resolution melt product will give strength to the results. In the revised manuscript we report the comparison with the 25 km dataset. We add the major results of the comparison (in terms of commission/omission error, melt extent estimation and melting trends) in the main paper and additional figures and tables in the supplementary material. For 25 km resolution data: https://doi.org/10.5067/MEASURES/CRYOSPHERE/nsidc-0533.001.

Also, I would like to suggest that the data and methods section (Sect. 2) is a mix-up of data, methods, results, and discussion, which confuses readers. Figures 2, 3, and 4, as well as Tables 3 and 4 should be presented in the results and discussion section. Please reformulate the section.

R: We agree with the reviewer. We divided Section 3 (Intersensor calibration) in two parts: 1) "methods" in Section 2.5 and 2) "results" in Section 3.1. Now, Figures 3 and 4 as well as Tables 3 and 4 are in Results and Discussion section. We left Figure 2 in PMW data description, as it is used as support to describe the differences in time-series between the new and the old datasets, providing a first picture of the differences between 25 km and 3.125 km at point scale. Moreover, in accordance with this suggestion, we moved the description of the methodology adopted in the semi-variogram analysis in a dedicated subsection in Data and methods section.

I would like to suggest that the authors should attend to the above-mentioned major

issues before considering its publication.

Other specific comments are as follows:

Specific comments: L. 25: More detailed explanation of "local scale processes" is needed here.

R: In the dedicated subsection in the "Results" section we explain the possible local scale processes affecting the spatial autocorrelation of melting. See "...local processes that drive melting as the melting season progresses (e.g., impact of bare ice exposure, cryoconite holes, new snowfall, etc.) and of a more developed network of surface meltwater, the presence of supraglacial lakes and, in general, the fact that the processes driving surface meltwater distribution (e.g., albedo, temperature) promote a stronger spatial dependency of meltwater production at smaller spatial scales."

L. 86 âĹij 90: It is necessary to introduce why such a high-resolution dataset from the Ka band product were not available until recently. What is the key innovation that enabled us to use the Ka band data for the detection of the ice sheet surface melt? It is also important to explain the difference in sensitivities of the K and Ka bands data to the liquid water clouds.

R: For what concerns the novelty of the high-resolution product and its recent availability, we described the improvements introduced in the gridding technique in subsection 2.1. We provide a description of the main steps and techniques adopted in the image reconstruction to reach the resolution of 3.125 km. We refer to this part as "More details are reported in the following sections". We better refer now modifying the statement in "More details are reported in Section2.1". We also added details about the coarse resolution "drop-in-the-bucket" technique, in order to make clearer the difference between the two products. For what concerns the difference between K and Ka bands, we thank the reviewer for pointing this out. The presence of the atmosphere is an important point to be taken into account when working with PMW spaceborne radiometers. The surface emission signal passes through the atmosphere and is affected

by its absorption and emission. The atmosphere affects the two frequencies of 19 and 37 GHz in a slightly different way. However, even if a difference exists, it is not that large and reduces as the brightness temperature increases (Tedesco and Wang, 2006). We expand this issue in the revised manuscript adding the following reference. Reference: Tedesco, M. and Wang, J. R.: Atmospheric correction of AMSR-E brightness temperatures for dry snow cover mapping," in IEEE Geoscience and Remote Sensing Letters, vol. 3, no. 3, pp. 320-324, July 2006, doi: 10.1109/LGRS.2006.871744.

L. 130: "2.2 Greenland air/surface temperature data": It is necessary to explain how the authors obtain surface temperature from the AWS (automated weather station) data. It is because the AWSs do not measure surface temperature directly.

R: Thank you for pointing out this. We corrected in "2.2 Greenland air temperature data". In the data description we actually refer to the data as air temperature (3m above the surface).

L. 132 âĹij 133: Strictly speaking, even if the surface temperature reaches 0 degreeC, it does not ensure that meltwater exits at the surface. How do the authors detect whether meltwater exits at the surface or not from the AWS data?

R: We used the air temperature from the automated weather stations available as a proxy of the presence of surface melting as done in Tedesco (2009). We certainly are aware of the limitations of this approach (that surface melting is not regulated by the temperature only and that the air temperature does not necessarily represent the snow surface temperature), however we classify a day as melting when the air temperature reaches the value of $0°C$ during the day. Moreover, we performed a sensitivity analysis considering as threshold values for air temperature the values of -1$°C$ and -2$°C$ in order to include possible melt events occurring at sub-zero air temperature conditions. Additionally, we performed the same commission/omission error analysis using the outputs of the regional climate model MAR. The use of MAR simulated LWC gives more robustness to the results obtained with the AWS analysis.

L. 185 âĹij 186: "Building on Tedesco (2009), we considered the two LWC values of 0.1% and 0.2 %": Please explain more in detail about this process. It is unclear why 0.1 and 0.2% are chosen here. L. 193: For MEMLS, why do the authors consider only the case of 0.2% LWC?

R: To respond to the last two comments, the choice of 0.2% of LWC is related to the rationale behind MEMLS algorithm, designed to detect small presence of liquid water (such as 0.2%). This algorithm is supposed to detect the sporadic melt events. We based our choice selecting the 0.2% according to the results presented by Tedesco (2009), cited in this paper, who tested both 0.2% and 0.1% liquid water content. Accordingly, the value of LWC=0.2% for the MEMLS algorithm better matches the number of melting days detected from other sensors (e.g. QuickSCAT). Contrarily, melting was overestimated by applying the algorithm based on 0.1%. To make the manuscript clearer to the reader, we remove the statements related to the 0.1% LWC as we do not consider it in the following sections.

L. 193 âĹij 196: "As we explain below, this choice was driven by the performance of the different considered algorithms. Moreover, we found that the fixed-threshold algorithm is more sensitive to persistent melting where the MEMLS-based one can detect sporadic melting. This allows us to analyze both melting conditions (sporadic vs. persistent) and analyze them within the long-term, large spatial scales that the PMW data can provide.": I think it is not necessary to state them here. They can be removed.

R: Removed

L. 304: "with the MEMLS being the most sensitive": The authors' intention is unclear. Sensitive to what?

R: Corrected as "with MEMLES providing the lowest threshold"

Technical corrections:

L. 16: "MeASUREs": Its definition should be indicated here.

R: Corrected indicating the definition.

L. 17: "Km" -> "km"

R: Corrected

L. 19: "MEMLS model": Brief explanation of the model or the definition of the abbreviation should be indicated here.

R: Corrected by indicating the abbreviation of the definition

L. 82: Please provide the definition of the abbreviation "rSIR".

R: Corrected

L. 103: "SMMR": Please provide its definition here.

R: Corrected

L. 131: "In order, to" -> "In order to"

R: Corrected

L. 146 âĹ̇ij 147: "Lateral and lower boundary conditions are prescribed from reanalysis datasets." -> "Lateral and lower boundary conditions of the atmosphere are prescribed from reanalysis datasets."

R: Corrected

L. 153: "meltwater extent" -> "melt extent"

R: Corrected

L. 195: "where" -> "whereas"

R: Corrected

L. 211 âĹij 214: Please follow the instruction how to indicate date and time in the text. https://www.the-cryosphere.net/submission.html#math

R: Corrected

L. 576 âĹij 579: Brodzik et al. is updated in 2020.

R: Corrected

References: Mote, T. L.: Greenland surface melt trends 1973–2007: evidence of a large increase in 2007, Geophys. Res. Lett., 34, L22507, https://doi.org/10.1029/2007GL031976, 2007. R: Inserted in References

---

## Author Comment (AC2) · 24 Jan 2021

Dear reviewer,

Thank you for your useful comments and suggestions. See below our reply.

P. Colosio, M. Tedesco, X. Fettweis and R. Ranzi
The authors present analysis using a new, higher resolution passive microwave dataset for determining surface melt across the Greenland Ice Sheet. The authors make a strong case for why such a dataset is important for monitoring the ice sheet and demonstrate that the higher resolution data allows us to study surface melt in greater detail, altering the magnitude of some of the temporal trends and providing sufficient resolution for more thorough spatial analyses. The work is novel, presenting a new dataset and analyzing it with an existing algorithm to study trends in surface melt extent and timing. The methods implemented are appropriate and sufficiently explained in most cases. In my specific comments, I have a few points that I would like to see addressed in terms of articulating implications of some of the issues the authors note with the data (i.e. differences in the four PMW sensors used, issue of poor matching between MEMLS and MAR5cm before 1992, MEMLS algorithm issues after main melt season). I do not consider any of these to be major issues; I would just like to see some clarification and explanation of the potential effects of this issues on the results. Additionally, an overall comparison of how this PMW melt detection compares to other PMW melt products in terms of commission and omission errors should be included in order to put this work into context. The results are significant, demonstrating that trends in surface melting are sensitive to the scale at which they are studied. The trends identified are important in our assessment of surface processes that affect mass balance and sea level rise. The surface melt product is an important dataset that can be used in future work as described in the conclusions.

The manuscript is overall well-written and flows logically, with only minor issues that will be easy to fix. Please find my specific comments and technical corrections below.

Specific Comments

line 171: Please explain what sigma is. It "varies in space and time" based on what –

is it the standard deviation?

R: Sigma is the standard deviation of the timeseries of brightness temperature for a specific year and pixel. We added this information in brackets in the revised manuscript.

line 190: I appreciate that many melt threshold/algorithms are implemented (and that they are compared to both in-situ data and the MAR output). Please explain why you selected the threshold/algorithms as you did given that you also presented at least 2 others.

R: Explained at the end of section 2.4 as "We selected M+$\Delta$T and MEMLS due to their higher accuracy in detecting both sporadic and persistent melting with respect to the other approaches presented above (i.e. Torinesi et al. (2003), Ashcraft and Long (2006) and MEMLS in case of LWC=0.1%) proved in previous studies (Tedesco, 2009). We selected also the 245K to test a more conservative approach aimed to detect persistent melting only."

line 247: With respect to the differences in acquisition time, is there a consistent lead/lag between timing of SMMR and SSM/I-F08? If so, how might the directionality of the lead/lag impact the analysis?

R: The lead/lag can be obtained by Table 2 where sensors characteristics are detailed. Specifically, in case of SMMR and SSM/I-F08, the lag of SMMR sensor is of about 6 hours (24:00 vs 18:17 for the ascending pass and 12:00 vs 6:10 for the descending pass). This constant lag can lead to errors and biases in particular at the beginning of the melting season when snow undergoes freeze/thaw cycles during the day (e.g. frozen snow, i.e. low Tb, at 6:10 for SSM/I-F08 and liquid water, i.e. high Tb, at 12:00 for SMMR in case of descending pass, the opposite in case of ascending pass). A possible consequence could be an early estimation of MOD from SMMR data (as already pointed out in Tedesco et al., 2009). In the revised manuscript we added "Specifically, we expect larger errors at the beginning of the melting season when snow undergoes thawing/refreezing cycles during the day, potentially having frozen snow

(low values of Tb) early in the morning and late at night (SMMR ascending and SSMI/-F08 descending passes) and presence of liquid water (high values of Tb) during the day."

line 260: Which correction did you apply to the SMMR data (the first method with weighted values or the second method using all values and the least square fitting) and why?

R: Thank you for noticing this. We specified it as "We applied the correction coefficients obtained with the second method according to the higher relative improvement for the evening pass."

line 260: What are the implications (if any) of not correcting the datasets? For instance, the average differences from F08-F11 and F13-F17 are positive, while the difference for F11-F13 is negative in the evening and positive (close to zero) in the morning. If agreement is worse when corrected, I agree that it makes sense not to implement the linear corrections, but it would be important to address what the potential effect of this is.

R: Possible implications in not correcting the dataset are related to the relative difference of the measurements from different satellites. This can cause errors in melt detection when considering the fixed threshold case (as 245K) but not in case of MEMLS which is computed considering intrinsic characteristics of the timeseries every year (i.e. winter average brightness temperature). However, the computed average difference of Tb in case of F08-F11, F11-F13 and F13-F17 is at most 0.52K, negligible with respect to the increase of Tb due to LWC.

line 306: Is there an emissivity threshold being considered here to indicate if melting is or is not occurring? If so, please add that.

R: Even if a rough threshold could be assigned (e.g. around 0.85), in this case we do not give a threshold. Instead the comparison is between the three computed values

of emissivity only. Considering a surely dry condition emissivity (0.74 in Figure 5a) and a surely wet snow condition emissivity (0.9 in Figure 5c), if melting occurred in the period 17 June and 17 July we would expect at least a value between case (a) and (c) (between 0.74 and 0.9). This happens in late July, when the brightness temperature "jump" is strong and evident, the air temperature reaches the melting threshold and, consequently, the emissivity reaches a value even higher than 0.9.

line 307: I think it was meant to say lower than in Summit Camp case?

R: Exactly, corrected.

line 315: (AWS Comparison Section) Were there any temporal trends the commission/omission errors of the melt algorithms as compared to the AWS data?

R: We thank the reviewer for this interesting question. We performed the comparison with AWS and MAR data to assess the different algorithms and select the best one, following Tedesco (2009) cited in this paper. We did not look at the temporal variability and trends of the commission/omission errors. We assumed that, if a trend does exist, it would have affected every algorithm. Thus, for our purpose, we only considered the overall error for every available year.

lines 325-327: I think the numbers for LWC1m and LWC5cm were swapped here?

R: Corrected.

lines 346-346: You bring up a very interesting point here. Because the brightness temperature after the largest part of the melt season has ended up lower than the Jan/Feb average, then the MEMLS algorithm would be less able to detect subsequent melt events. Is this a consistent pattern that is observed across sites/years? This could lead to a change in the frequency of omission errors of the MEMLS algorithm pre and post main melt season. Please discuss potential implications of this issue.

R: We found this pattern at Swiss Camp site at multiple years as it is possible to see in Figure 5c where the timeseries of Tb in 2006 is reported. The 2006 example seems

to confirm the hypothesis: after the main melting season (mid-august, between day 200 and 300) the Tb drops to values lower than before the melting season. After day 300 another jump of the signal is detected (for 1 day only, by MEMLS and the M+DT) followed by a further decrease of Tb. In this case the melt event is detected. Consequently, the lowered capability of MEMLS to detect melting is not a constant issue and it does not necessarily affect the omission error significantly. Similarly, before the melting season a sporadic melt event is detected (before day 100 for 1 day only, by MEMLS and M+DT), followed by a drop of Tb to values lower than before the melt event. It would be possible to expand this interesting point in another research work, addressing the causes and implications of these early/late sporadic melt events.

line 355: How do the commission and omission errors for these algorithms compare to other PMW melt detection products.

R: We compared the commission and omission errors presented in the submitted version of the manuscript with the ones obtained for the Thomas Mote 25km 19 GHz PMW dataset suggested by the other reviewer. We include the averaged results in Table 5 in the revised manuscript. This PMW dataset is at the resolution of 25 km and uses the 19 GHz frequency, enabling the comparison with a coarser resolution data and giving us the possibility to show the benefits of the highest resolution. We found that, on average, omission and commission errors are lower in case of the higher resolution dataset. Moreover, the comparison with MAR 6km outputs shows lower NSE values in case of the 25 km 19 GHz PMW dataset.

lines 369-371: Please explain your decision to compare 245K and MAR1m and also compare MEMLS and MAR5cm. I believe it is because the expected differences in sensitivity of each of the different methods of detecting melt, but just want to be sure that is why this decision was made.

R: We added an explanation of this choice as "due to the expected differences in sensitivity to detect persistent and sporadic melting between 245K and MEMLS, respectively"

lines 375-380: It seems that using the SMMR data (from 1979-1987) is part of the issue here. Is that correct? Is it partly because of the difference in time of day? Or difference in sensor technology used?

R: It seems that the main issue is related to the different sensor technology. Even if we improved the consistency of the timeseries by calibrating the SMMR data, differences still remain, partly because of the different acquisition time and frequency and partly because of the specific characteristics of the sensor (e.g. different IFOV, swath width, incidence angle). Added a sentence: "(…) possibly due to a persistent bias after the intercalibration of the dataset. (…)"

line 382: What are the implications of the melt extent being underestimated?

R: Added "A possible consequence of the melt extent being underestimated in the first part of the timeseries is a slightly overestimated long-term trend. To address this possible implication, in the next section we compute long-term trends considering both 1979 – 2019 and 1987 – 2019 reference periods."

lines 396-397: Is there precedent for using this definition of MOD and MED?

R: Following Tedesco et al. (2009), cited in this paper, we defined MOD and MED as the first and the last two days in a row when melting occurs. Tedesco et al. (2009) identified the first and last days as MOD and Med using a double condition algorithm. Here chose to consider two consecutive days as we prescribe a single melting condition (Tb>threshold).

line 406: Here and elsewhere you refer to the trends in the coarser-resolution data. Please consider including this analysis in supplemental material.

R: In the revised manuscript we will substitute this analysis with the comparison of the trends computed using the Mote PMW dataset, reporting the analysis in the supplemental material.

line 436: Is there any explanation for the areas in the map with anomalous trends? (figure f, negative trend in Northern Greenland, figure d, positive trend in some regions in central Greenland)

R: A possible explanation can be related to the definition of MOD and MED (first two consecutive days when melt occurs and stops). Possibly, by modifying the constrain of two consecutive days (e.g. a single day or even 3 or 4 consecutive days) the anomalous areas would reduce. On the other hand the parameter melt duration MD is more spatially continuous in trend evaluation.

lines 436-438: How are pixels that do not consistently (every year) experience melt handled?

R: In case of pixels that do not consistently experience melt, when computing the pixel-scale trends for MOD and MED, we performed the calculations for the available data only. In case of melt duration (number of days detected as melting for each pixel), instead, we consider as MD=0 in case of a pixel presenting zero melting days.

lines 440-446: This content reads more like methods. Consider relocating the description of the methods of the semi-variogram analysis.

R: We moved the description of the semi-variogram analysis in a new sub section "2.6 Spatial autocorrelation: the variogram analysis" where we describe the methodology adopted.

line 452-453: The comment about extending this analysis seems out of place.

R: Removed.

line 457: Consider showing the figures that accompany the data for Table 7. Perhaps in the supplement at least?

R: In the revised manuscript we report the figures asked in the supplement.

line 458: These are semi-variograms for melt duration in each month. Is this the num-

ber of days of the month that melt occurs for a given pixel? Do the days need to be consecutive? I think more detail about the melt duration variable should be provided here.

R: Described adding the following sentence: "Here, we compute the melt duration for each month of the melting season at pixel-scale as the number of days of the month (May, June, July or August) detected as melting for the specific pixel."

lines 464-468: Is there a way to compute uncertainty associated with these distances?

R: It could be possible to evaluate the variability of these distances by performing a larger analysis for every year of the timeseries (1979-2019). We are considering to expand this aspect in a future research focused on this aspect.

lines 475-476: What does the larger nugget value for MAR as compared to the PMW data tell us?

R: We think that the difference in nugget value is mainly related to the different spatial resolution of the considered datasets. The nugget effect is affected by the volume of sampling, decreasing in value as the volume increase. The nugget effect can be attributed to measurement errors or spatial sources of variation at distances smaller than the sampling interval or both. Measurement error occurs because of the error inherent in measuring devices. Natural phenomena can vary spatially over a range of scales. It is difficult to say what drives this difference without in-situ data (both melting and passive microwave). We note that this does not impact the results on the scale break properties.

lines 499-500: The sentence about the threshold for melting seems out of place in the conclusions.

R: Removed

line 500-501: The data do not seem to support "good matching" in most of the years from 1979-2019. The data do seem to support good matching from 1992-2019. Please

add this caveat to the statement.

R: Corrected as: "We obtained good matching (i.e., NSE>0.4 or, at least, positive) in most of the years from 1992-2019 when comparing MEMLS derived melt extent with MAR liquid water content in the first 5 cm of snowpack. On the other hand, we found bad matching in the period 1979-1992, possibly due to differences in sensor characteristics."

Technical Corrections

line 11: modulation "of" ice dynamics

R: Done

line 13: "in view of" should perhaps be replaced with "due to"?

R: Done

line 17: km instead of Km

R: Done

line 19: capable "of detecting"

R: Done

line 25: the word "interest" seems out of place. Delete or replace, perhaps with "usefulness"?

R: Deleted

line 26: monitor should be "monitoring"

R: Done

line 74: here and elsewhere you use Tbs, when I think leaving it singular as Tb is more clear

R: Thank you, corrected

line 188: Should this be Tc or Tb? If it is meant to say TÂnc, please define this term. ËǦ

R: We use Tc to refer to the threshold brightness temperature value. We added the definition where first introduced as "(. . .)Tc indicates the threshold value (we keep the same notation in the following)"

line 190: "as sensitivity to Zwally. . ." not sure what is meant here. Typo?

R: Zwally and Fiegles (1994) proposed the DT=30K. Here we test DT=35 K and DT=40 K to test the sensitivity of the algorithm selected. I correct the statement expanding it as ". . .equal to 30K and, to test the sensitivity to Zwally and Fiegles (1994), 35K and 40K (M+30, M+35 and M+40 from here on). . ."

line 251: R2 needs to be a superscript 2

R: Done

line 255: specify that you referring to data in Table 4 here

R: We are actually referring to Table 3 here, specified.

line 267: move "daily averaged from AWS" to directly after air/surface temperature to improve sentence clarity

R: Done

line 281: correct 919% to 9.19%

R: Done

line 305: fix subscript on Tb

R: Done

line 336: Is this average surface air temperature? Please specify.

[Figure]

R: Yes it is, specified

line 342: detected by the threshold algorithms in AWS temperature? By all three?

R: Thank you for noticing this, only for Tair=-1°C and -2°C. We added "(Tair>-1°C)"

line 344: should be "corresponds"

R: Corrected

line 355: consider describing it not as an overall error but as what it is a mean of errors calculated using different techniques.

R: Substituted "overall" with "average"

line 364: perhaps rephrase as "Here, we remind the reader that.."

R: Done

lines 399-400: typo? Partial repeating of a line

R: Yes, typo. Corrected

lines 405-406: typo in years indicated here?

R: Yes, substituted 2016 with 2019

line 418: Fix figure numbers

R: Corrected to Figure10d

line 468: some missing words, should read "the value of the range results is lower in the case of.."

R: Corrected as "(…) the value of the range is lower (…)"

line 473: "till" should be "until"

R: Corrected

line 503: typo of word largely

R: Corrected

line 525: Perhaps add "We have" to "assessed the capability. . .."

R: I think it is correct as it is.

line 729: (Figure 1) consider including scale bar for figures c and d

R: It is the same scale bar of Figures a and b.

line 739: (Table 1) table caption perhaps should say "of the selected Greenland Climate Network (GC-Net) sites"

R: Corrected

line 752: (Table 4) This table shows regression analysis for more comparisons than just SMMR and SSM/I-F08. Please update caption to reflect this.

R: Corrected substituting with "between the selected couples of satellites"

line 760: (Figure 4) Please ensure that y-axis are the same for all three panels of figure 4a

R: Corrected. I also increased the thickness of the lines and the colors to make the figure more readable.

line 771: (Figure 6) Please consider adding labels to each map for ease of interpreting the figure

R: We apologize but we do not understand this specific question. However, we reported in the captions all the information needed to interpret the figure.

line 780: (Figure 7) Please indicate what the vertical teal lines represent

R: Done

---

## Author Response (AR2)

**Dear Editor,**

**Thank you for reading our manuscript and providing your comments. In general, we tried to increase the consistency in the use of the acronyms as requested to improve readability of the manuscript.**

**Please, find below our response to the specific comments.**

*Dear authors,*

*This manuscript was reviewed by two referees who requested a major revision. The revised version was reviewed by one referee who judged that the revised version addressed their concerns adequately. However, I would like to request another revision to improve clarity and readability of the manuscript. Surface melt of Greenland is a highly interesting topic for a broad range of scientists, and satellite remote sensing examined in this manuscript is one of the key observational tools over the entire ice sheet for a historic period with a high temporal resolutions.*

*I would like to use Table 5 as an example to explain my point. This table includes M (most left column), T, LWC1m and LWC5 cm (second column from left), and C+O on the top row. This table is cited at p.12, whereas M is defined as the January-February mean brightness temperature at p.6 (why "M", by the way, and is it different from Twinter) and it is not cited in the main text where Table 5 is cited. T is not defined, and AWS-measured temperature is defined as Tair at P12L356. LWC is not defined but used many times. I guess "C+O" refers "total error including commission and omission" but it is not defined in the manuscript. I would add headers such as "satellite threshold" and "AWS threshold". In this way, even if M for example is not defined in the text where this table is cited, the readers can develop good understanding of this table. And I would expand the caption to something like "Performance of surface melt detection examined with AWS data. Six thresholds (rows) are used to determine satellite-detected surface melt. For each case, algorithm performance was examined using AWS data with five thresholds to determine surface melt. Commission and omission errors are shown in percentage of the total number of days considered averaged over all the selected AWS. [add explanations for C+O and C+O mean]" For the broad TC's readership, this level of description is necessary.*

*I echo reviewers' view; this manuscript presents highly interesting and valuable science. However, my own evaluation is that readability and clarify of this manuscript should be improved otherwise the broad readership of TC cannot get the valuable science that you are presenting here.*

*In general, I request following changes.*

*1. Remove most of acronyms that do not use so often. Such terms include GrIS, SLR, CDRs, MRF, T-B, E-D, and LTOD (this is not a comprehensive list, and I request the authors to check the entire manuscript carefully).*

**R: Thank you for pointing this out. I read through the whole manuscript and removed the least relevant and not often used acronyms (all the acronyms mentioned here and others, e.g. SISVAT, E and M for evening and morning).**

*2. When defined, use acronyms consistently and do not define other acronyms for the same term. Such terms include MAR5cm vs. LWC5cm, M vs. Twinter, and Tair vs T. These terms may refer different values; if it is the case define them more clearly and use them consistently.*

**R: Thank you for this important comment. I checked the manuscript and corrected all the double acronyms referring to the same term. This should improve the readability of the manuscript.**

*3. Add definitions of all acronyms. For example, LWC, C+O.*

**R: Checked the acronyms and added the definitions.**

*4. Section 3.3 define many acronyms, such as MD, MMD, MMS, MI, MOD, MED, and MMS. If these acronyms are widely used in passive microwave remote sensing and use of these acronyms improves the clarify of this paper for specialists, you can keep using them in the tables and figures, but define them in the captions and minimize their use in the main text.*

**R: Yes, these acronyms are widely used in passive microwave remote sensing of surface melting and I think that using them is important to keep the same notation of previous works. I defined them in the captions (specifically Figures 9 and 10).**

*I made some notes below, but please carefully check the entire manuscript to fully improve the manuscript in this round. I expect comprehensive revision to improve readability, and just responding to following specific points does not constitute adequate revision that warrants the acceptance of this manuscript.*

*All page/line numbers refer those in the mark-up version.*

*Title: change to "derived from"*

**R: Agree, changed**

*P1L12 and L14: do you need to define PMW? Is it better to define PMW as passive microwave at 37GHz vertical polarization?*

**R: I would prefer to keep this acronym since it has been historically used in the passive microwave notation.**

*P1L22-L24: present these numbers in a consistent manner. If you indicate the number of days to the first decimal point (e.g. 2.5 days), please do so for the both cases, instead of about 4 days and 2.5 days.*

**R: Corrected. Thanks**

*P2L2: Unit for the ice volume in Greenland is km3, not m3.*

**R: Thank you, it was a typo. Corrected.**

*P2: Many acronyms are defined in an inefficient manner in this paper. Defined acronyms in the first few pages are GrIS, SLR, SMB, PMW,~rSIR, CDRs,~MRF, GC-Net. Please critically review the manuscript and remove unnecessary acronyms. For example, I don't think following acronyms are necessary: GrIS, SLR, CDRs. Also, once it is defined please use it consistently. I see many sentences that do not use these acronyms (e.g. P3L94, P3L97, P5L141).*

**R: Thank you for this precious suggestion. I agree that some of the acronyms are not necessary. I removed GrIS, SLR, rSIR, SMB, MRF. I would like to keep PMW and Tb as separate acronyms, as generally done in passive microwave literature. Also, the main melting parameters, widely used in literature, are left in the manuscript (MMD, MOD, MED etc.). I checked the entire manuscript in order to make a better and consistent use of them.**

*P3L75: Spellout MeASURes (like you did in Abstract).*

**R: Corrected**

*P3L94-95: Revise. I cannot get the point.*

**R: Revised as "However, when considering higher values of $T_b$, the difference of the atmospheric effect between 37 GHz and 19 GHz $T_b$ decreases (Tedesco and Wang, 2009)."**

*P4L100: Does "estimates of surface melting" refer melting timing, duration and/or amount?*

**R: Specified as "We compare results from these algorithms with estimates of surface melting obtained from data collected by automatic weather stations (AWS) in terms of melting timing and with the outputs of the regional climate model Modèle Atmosphérique Régional (MAR; Fettweis et al., 2017) in terms of melting timing and extent."**

*P5L130: remove coordinates of the Swiss Camp. It appears here and also Fig. 2 caption, while it is shown in Table 1.*

**R: Removed.**

*P6L174: LWC is not defined*

**R: Defined in subsection 2.3 as: "…we average the liquid water content (LWC) simulated by MAR along the vertical profile..."**

*P6L177: T-B and E-D are defined but not used. I don't think you need to define them.*

**R: Removed**

*P6L189: What's the difference between M and Twinter? Also, Twinter is defined so please use it consistently (e.g. P7L198 says winter Tb).*

**R: It's the same, I corrected the manuscript keeping everywhere the same notation. I choose to keep M to identify the average winter brightness temperature as done in Tedesco (2009) (cited in this paper).**

*P7L214: MEMLS refers the microwave emission model of layered snowpack in general, so it is confusing if MEMLS is used in two different ways. Indeed, the authors write MEMLS (LWC = 0.2%) later, which is much better to refer MEMLS for a specific case. However, P7L219 uses MEMLS(0.2% LWC). Again, please use these terms consistently. The current manuscript is very hard to read and consistent use of terms will improve readability significantly.*

**R: Since the only specific case of MEMLS presented is the one with 0.2% of LWC, we refer to the algorithm simply as MEMLS. I corrected the double/inconsistent notetions and specified: "…the algorithm based on MEMLS in case of LWC=0.2% (referred simply as MEMLS from here on for brevity) and the 245 K fixed threshold (245K from here on)" to improve clarity.**

*P8L224 and P8L234: What do you mean with "novel nature" and "novelty"?*

**R: It refers to the enhancement in spatial resolution, a crucial novelty in passive microwave remote sensing. We specify now in the text as: "In view of the novelty of this PMW dataset introduced by the enhancement in spatial resolution thanks to the improvement of the gridding technique, .."**

*P12L367, P12L373, P13L380: air temperature or surface temperature?*

**R: Air temperature. Corrected.**

*P13L385 and L387: It was defined as Delta T, not Delta T_b (P6L189).*

*(I stop commenting editorial issues in the main text here, but request the authors to carefully review the current manuscript and make serious efforts to increase readability of the manuscript).*

**R: Corrected.**

*Table 1: change to "…. Used to validate the results in this study." If the table includes the sites not used in this work, remove such sites.*

**R: Corrected.**

*Table 2: Do you mean " the sensors used for this work."? "Selected" can be interpreted in many ways and thus confusing. Spell out IFOV in the table or at least define it in the caption.*

**R: Corrected as: "Instantaneous Field of View (IFOV) [km2]". I removed "Ka-band" in brackets since the frequency is already defined in another table line.**

*Figure 2: Add Tb after "Brightness temperature" in the y axis label in panel a. The label April 3rd 2012 points to more than 10 days. Use an arrow, instead of circle to specify the date. ⬚ and ⬚ are not used often, so I suggest removing them and spell out mean and standard deviation every time. If it is the case, remove these values from the panel and add them to the caption. Remove Swiss Camp coordinates.*

**R: I substituted brightness temperature with Tb, as it is an acronym already defined. Substituted the circle with the arrow and moved mean and standard deviation information in the caption. Removed Swiss Camp coordinates.**

*Figure 3: Is the data density (colorbar) normalized? A clarification is needed, what does the value 1 mean?*

**R: it is the relative frequency. Added in the caption.**

*Table 3: the left top cell says platforms, but they are sensors (keep Tables 2 and 3 consistent). Spell out PMW (or at least maintain the same style whether caption uses acronyms or spell-outed full terms). Clarify whether "the difference" is $T_{(high-ress)} - T_{(low-ress)}$ or vice versa.*

**R: Thank you, it's sensors. Corrected. The difference is indicated in the "Senrsors" (e.g. SMMR – SSMIF08)**

*Figure 4: clarify "Distance" in the caption.*

**R: Done.**

*Figure 5: M+DT should use symbol D. Do you need define epsilon to refer emissivity here? Also it is confusing that epsilon is used to define emissivity and the label has the upper bar to show that it is a mean value. Please minimize the use of acronyms. Remove one unnecessary arrow in panel a.*

**R: Corrected the DT. Removed the additional arrow. Corrected the upper bar epsilon in simply epsilon, typically used in passive microwave notation to denote the emissivity.**

*Figure 7: Add labels to all y axis. The labels show LWC in percent. If so, it is better to show y axis in percent too.*

**R: Corrected.**

*Table 6: What is "245K (MEMLS)" algorithm? Also, the table includes MAR1m MAR5cm, which should be changed to clearer acronyms. They are explained in the caption, but please use the acronyms consistently throughout the paper.*

**R: Corrected. Also added a sentence referring to the coarse resolution data (reported in table). I preferred to leave MAR5cm and MAR1m as acronyms, clearly defining in the text and using them consistently. In particular I removed the acronyms LWC1m and LWC5cm. I think that now the manuscript is more readable.**

*Figure 8: check the year collected these data. Caption says 1984 and 2006, but the table labels say 1983 and2005.*

**R: Yes, thanks. It's 1983 and 2005. Corrected.**

*Figure 9: Explain solid and dashed lines in the panel. Also consider spelling out all y label terms, or at least define them in the caption. Does panel b show the values in percentage? If so, clarify it in the y axis label.*

**R: The difference between solid and dashed lines is explained in the caption. I defined MMD, MMS (specifying that it is expressed as fraction of the surface area of the ice sheet), MI, MOD and MED in the caption.**

*The manuscript includes supplemental materials (5 figures and 4 tables), but none of them is cited in the main text. I have no objection to include them as supplement in this paper (if it is usefully cited), but it is also possible to archive a dataset reporting the values in these tables as well as the those reported in tables in the main text at a data center such as NSIDC or Pangaea.*

**R: I would like to keep the supplementary material. I think it can be interesting for the reader to have, for example, a graphical comparison for the coarse resolution dataset that might be on the other hand unnecessary in the main text. I added the reference to each supplementary figure in the main text.**

---

## Author Response (AR3)

Dear Editor,

Thank you for reading our manuscript. We completed the required technical corrections, all of them were crystal clear. Here the point-by-point response to your suggestions.

Paolo Colosio

P4L111: Change the current Table 2 to Table 1 and cite the new Table 1 here.

R: Done

P5L130: remove blue and red (explained in the figure caption, which is enough)

R: Done

P5L134: remove the reference to Figure 2.

R: Done

P5L150: Because of swapping the order of Table 1 and Table 2, here it should be "… in Table 2".

R: Done

P5L157: non-capitalize Soil/Ice Vegetation Atmosphere Transfer.

R: Done

P6L169: "…. Along the vertical profile of snow from the surface to 5 cm or to 1 m"

R: Corrected as "…along the first 5 cm and 1 m from the surface of the vertical profile of the snowpack"

P6L191: "LWC, snow density"

R: Corrected

P7L221: Change Table 2 to Table 1

R: Done

P11L323: "the sum of the two Commission and Omission errors"

R: Done

P11L331: "0oC"

R: Done

P13 – P14: Improve clarify. I think NSE values (L416), values of NSE (L417), Nash-Sutcliffe Efficiency coefficient (L444) refer the same value and then it is always write it down as NSE coefficient.

R: Corrected, now it is always written as "NSE coefficient"

P14L421: remove "(ME)"

R: Removed

P14L441: apostrophe after timeseries?

R: Removed

P14L444: change Nash-Sutcliffe Efficiency to NSE

R: Changed

P15L461: remove "(MOD)"

R: Removed

P15L463: Supplemental figures S2, S3, S4 and S5 are cited altogether. Please cite individual supplemental figures, with a short description what each figure presents.

R: Cited individually when the results are reported.

P16L488-489: is it in Figure 9d or Figure 11?

R: Figure 9d, corrected

P16L493: Change to "MEMLS between 1979-2017 …. And MEMLS between 1998-2012"

R: Changed

P17L526: change Figure 12 to Figure 11 (there is no Fig. 12)

R: Thanks, corrected

P17L531: Change Fig. 12 to Fig. 11

R: Corrected

P17L545 cite Fig. S5 after "in August"

R: cited

P18L547 and L549: does "autocorrelation" refer the range specifically or more generally the autocorrelation features? If the former is the case, please change it to range.

R: hank you. L547 refers generally to autocorrelation features, L549 to the range we computed. Corrected.

P18L573: change to LWC

R: Done

P18L562ff: it is confusing to see enhanced dataset, high resolution dataset etc in this section. Define the term at L532 (e.g. "3.125 km resolution 37 GHz horizontal polarization PMW Tb, called higher-resolution dataset") and use this term consistently.

R: Corrected consistently using "PMW enhanced-resolution data"

P19L581-582 and 591, 595: reverse changes to MOD, MED, MMD, MMS and MD (some readers read conclusions first and extensive use of these terms does not help such readers).

R: corrected using melt onset date, melt end date, mean melt duration, maximum melting surface and melt duration.

Table 1 and Table 2: swap the order.

R: Done

Figure 2 caption: at the very ends "mean equals …. Standard deviation equals to…."

R: Corrected

Table 3: in the top row, change "SMNR vs F08", "F11 vs. F13 (Evening)" or such like Table 4.

R: Changed

Figure 4: Add the unit of distance (m?). Also, change first, second and third rows to left, center and right.

R: Changed from first, second and third to left, central and right. The "distance" is unitles as it is the difference between the two histograms (black and purple).

Table 6 caption: change to "Nash-Sutcliffe Efficienty (NSE) coefficients" and change MOTE to Mote, 2014.

R: Corrected.

Figure S5: add "Table 7 reports range, sill, nugget and R2 values of these semi variograms."

R: Added.